# Lethal Borna disease virus 1 infections of humans and animals – in-depth molecular epidemiology and phylogeography

Borna disease virus 1 (BoDV-1) is the causative agent of Borna disease, a fatal neurologic disorder of domestic mammals and humans, resulting from spill-over infection from its natural reservoir host, the bicolored white-toothed shrew (*Crocidura leucodon*). The known BoDV-1-endemic area is remarkably restricted to parts of Germany, Austria, Switzerland and Liechtenstein. To gain comprehensive data on its occurrence, we analysed diagnostic material from suspected BoDV-1-induced encephalitis cases based on clinical and/or histopathological diagnosis. BoDV-1 infection was confirmed by RT-qPCR in 207 domestic mammals, 28 humans and seven wild shrews. Thereby, this study markedly raises the number of published laboratory-confirmed human BoDV-1 infections and provides a first comprehensive summary. Generation of 136 new BoDV-1 genome sequences from animals and humans facilitated an in-depth phylogeographic analysis, allowing for the definition of risk areas for zoonotic BoDV-1 transmission and facilitating the assessment of geographical infection sources. Consistent with the low mobility of its reservoir host, BoDV-1 sequences showed a remarkable geographic association, with individual phylogenetic clades occupying distinct areas. The closest genetic relatives of most human-derived BoDV-1 sequences were located at distances of less than 40 km, indicating that spill-over transmission from the natural reservoir usually occurs in the patient´s home region.

Borna disease virus 1 (BoDV-1, species *Orthobornavirus bornaense*, family *Bornaviridae*) is the causative agent of Borna disease, a severe and often fatal neurologic disease of various domestic mammals, particularly horses, sheep, and New World camelids[1–5]. Recently, the virus received increased attention following confirmation of fatal BoDV-1-induced encephalitis in humans[6–18]. In domestic mammals and humans, BoDV-1 establishes a persistent and strictly neurotropic infection mainly of the central nervous system, with so far no evidence of shedding of infectious virus. The infection usually results in non-purulent encephalomyelitis due to T lymphocyte-mediated immuno-pathology with a high case fatality rate[1,7,9,19,20]. The incubation period is presumably highly variable and assumed to range from several weeks to a few months in most cases[8,21,22], which may at least partly be the result of different infectious doses or routes of infection[23]. Affected

individuals usually develop fever accompanied by headaches in humans, followed by a broad range of behavioural and neurologic disorders, including apathy, compulsive movements, seizures, ataxia or blindness. In most cases, the disease progresses to coma and death within days to months after the onset of neurological symptoms[1,2,6,7,10,17,24,25]. Several compounds have been identified to possess antiviral activity against orthobornaviruses in cell culture but no therapeutic regime for animals or humans has been established yet[26–29], though in some human cases intensive treatment attempts were made[15]. As licensed vaccines against BoDV-1 are likewise not available[30], prophylactic measures are limited to reducing exposure to the BoDV-1 reservoir.

The known natural reservoir host of BoDV-1 is the bicoloured white-toothed shrew (*Crocidura leucodon*), in which BoDV-1 is not

✉e-mail: Dennis.Rubbenstroth@fli.de

strictly neurotropic but also infects epithelial cells in various organs, thereby allowing for shedding of infectious virus via saliva, faeces, urine, and skin scales[4,31–34]. However, the routes of BoDV-1 transmission within shrew populations as well as for spill-over to domestic mammals and humans remain largely unknown[1,17,34,35]. The known distribution range of bicoloured white-toothed shrews covers large parts of the temperate zone of Europe and Western Asia, extending from the Atlantic ocean to the Caspian Sea[36,37]. Nevertheless, BoDV-1 appears to be prevalent only in comparably limited regions covering parts of Eastern and Southern Germany, Austria, Switzerland and the Principality of Liechtenstein, based on the occurrence of BoDV-1 infection in spill-over hosts[4,7,25,33,38]. Previous work had demonstrated BoDV-1 sequences to constitute four separate phylogenetic clusters (designated 1–4) with two subclusters (1A and 1B), which appear to be associated with different regions within the endemic area[4,7,8,11,12,25,33,35,38–41].

The aim of this study was to provide an in-depth analysis of the molecular epidemiology and phylogeography of BoDV-1, in particular its spatial distribution in Germany and neighbouring countries, and to assess geographic risk areas for potential spill-over transmission to domestic animals and humans. As the available data on the occurrence of BoDV-1 in shrew populations are highly fragmentary and strongly dependent on the activities of a small number of research groups[4,25,31–34,42], we collected material from archived and recent cases of Borna disease in domestic mammals, serving as indicators for the presence of the virus. Additional diagnostic samples were obtained from BoDV-1-infected human patients and from bicoloured white-toothed shrews collected in BoDV-1-endemic regions in Germany. BoDV-1 sequences were generated from this material and used for phylogeographic analysis including also sequence data derived from public databases.

## Results

### RT-qPCR confirmation of BoDV-1 infections in domestic mammals, humans and shrews

Veterinary and human pathologists and diagnostic laboratories submitted fresh-frozen or formalin-fixed paraffin-embedded (FFPE) brain tissue or cerebrospinal fluid (CSF) from 231 suspected BoDV-1 infections in domestic mammals, 29 humans and seven BoDV-1-positive shrews (Table 1). The animals originated mainly from the known endemic regions of Germany (Bavaria, BY; Saxony-Anhalt, ST; Saxony, SN; Brandenburg, BB), Switzerland (Grisons, GR), the Principality of Liechtenstein and Austria (Vorarlberg, VA), with few exceptions originating from regions in Germany and Switzerland not previously known to be endemic for BoDV-1 (Fig. 1A). Quantitative reverse transcription polymerase chain reaction (RT-qPCR) confirmed the BoDV-1 infection for 207 out of 231 domestic mammals (89.6%) and all analysed shrews. Of the 29 human cases analysed in this study, 28 (96.6%) could be confirmed by RT-qPCR. RT-qPCR as well as high-throughput sequencing (HTS) remained negative for FFPE brain sections from a previously published case from Lower Saxony (NI) in 1992[13], possibly due to low RNA quality resulting from long-term storage of the material.

Of the two BoDV-1-specific RT-qPCR assays used for screening, the matrix protein (M) gene-specific RT-qPCR BoDV-1 Mix-6 yielded significantly lower cycle of quantification (Cq) values for FFPE material than the phosphoprotein (P) gene-specific BoDV-1 Mix-1 ($P < 0.0001$; paired Student´s t-test), whereas Mix-1 achieved significantly lower Cq values for fresh-frozen samples ($P < 0.0001$; Supplementary Fig. 1). The apparently higher sensitivity of Mix-6 RT-qPCR for FFPE-derived RNA, as compared to Mix-1, is presumably due to its shorter amplicon (75 vs. 162 base pairs, bp), allowing for a more efficient detection of highly degraded RNA.

### BoDV-1 infections in domestic mammals and humans

We then analysed the metadata available for the cases confirmed in this study in combination with previously published BoDV-1 infections of domestic mammals, humans and shrews, for which at least the BoDV-1 genome sequence spanning the nucleoprotein (N), accessory protein X (X) and P genes (1824 nucleotides [nt]) were available. These cases represented 55 domestic mammals, 16 humans, 36 shrews and 3 laboratory isolates originating from horses (Table 1). Samples from most of the 207 confirmed BoDV-1 infections in domestic mammals were collected between 2000 and 2023, but individual cases dated back as far as 1964 (Supplementary Fig. 2A, B). The seasonal pattern of Borna disease in domestic mammals was analysed for all RT-qPCR-confirmed cases together with all previously published cases included in this study with available information on month of death (Table 1). Death of BoDV-1-infected animals peaked in May and June, whereas the lowest numbers were observed during September to November (Supplementary Fig. 3A).

This study raises the number of published laboratory-confirmed human BoDV-1 infections to 46 (Table 1), following the case definition of Eisermann et al.[11], which requires direct virus detection from the patient. Twenty-eight cases were confirmed by RT-qPCR during this study (including previously published cases without BoDV-1 sequence; refs. 7,9,14,16–18,43–45). BoDV-1 sequences were available from public databases for further 16 previously published human cases (Table 1)[6–8,10–13,15]. Despite the lack of direct virus detection, the donor and the surviving liver recipient of a previously published solid organ transplant cluster were likewise regarded as confirmed cases due to their seroconversion and their unequivocal link to the confirmed BoDV-1 infections in both kidney recipients[8].

The metadata assembled for all 46 patients (20 females and 26 males) revealed a diagnosis of fulminant encephalitis for 45 patients, with a fatal outcome in 44 of the encephalitic patients (97.8%). The only exceptions from these characteristics were the transplant donor and the liver recipient of the previously published solid organ transplant cluster. While the donor had died of an unknown cause without brain histopathology being performed, the liver recipient survived the acute encephalitis with severe sequelae[8]. The age of the patients ranged from 7 to 79 years (median 53.5 years; Supplementary Fig. 4A). The first of these confirmed cases occurred in 1996 and had been diagnosed retrospectively[10]. Six patients died in 2016, representing the highest number of confirmed non-transplant-derived cases per year (Supplementary Fig. 2C). The highest number of deaths was recorded in November (Supplementary Fig. 3B), while the highest number of hospitalisations was reported in May (Supplementary Fig. 3C). The median time from hospital admission to death was 29 days (range: 4 to 274 days; Supplementary Fig. 4B).

### Phylogenetic analysis identifies a novel BoDV-1 cluster 5

Sequencing of the BoDV-1 genome was attempted using HTS and/or Sanger sequencing for 157 cases, including 120 domestic mammals, all RT-qPCR-positive humans ($n = 28$) and shrews ($n = 7$) and the laboratory isolates H24 and DessauVac (Table 1). Overall, BoDV-1 sequences covering the complete coding region of the genome ($n = 54$) or at least the N and X/P genes ($n = 82$) were successfully generated from 136 individuals, including 102 domestic mammals, 25 humans, all seven shrews and both laboratory strains (Table 1). Sequencing attempts failed or yielded only short sequence fragments for 18 domestic mammals and three human cases, mainly due to low viral loads and/or insufficient RNA quality in FFPE and CSF samples.

Including the newly generated BoDV-1 sequences as well as those derived from public databases (Table 1), two maximum likelihood (ML) trees were constructed for 90 complete coding BoDV-1 genomes (Fig. 2) or 246 N-X/P sequences (Fig. 3A and Supplementary Fig. 5). Both trees supported the previously published clusters 1 to 4 and the subclusters 1 A and 1B with high statistical support (SH-aLRT ≥80% and ultrafast bootstrap ≥95%). The sequences from two human cases from BY analysed in this study did not fall into any of the previously

**Table 1 | Numbers of cases and sequences included in this study**

| Parameter | Domestic mammals[a] | Humans | Shrews | Laboratory strains[b] | Total |
|---|---|---|---|---|---|
| **Cases analysed in this study** | 231 | 29[c] | 7 | 2 | 269 |
| Fresh or fresh-frozen samples | 48 | 14 | 7 | 2 | 71 |
| FFPE samples | 183 | 15 | 0 | 0 | 198 |
| Confirmed by RT-qPCR | 207 | 28 | 7 | n.a.[d] | 242 |
| Comparative Mix-1/-6 results | 204 | 23 | 7 | n.a. | 234 |
| Selected for sequencing | 120 | 28 | 7 | 2 | 157 |
| Sanger sequencing | 31 | 5 | 7 | 0 | 43 |
| High throughput sequencing | 89 | 23 | 0 | 2 | 114 |
| Bait-based enrichment | 14 | 2 | 0 | 0 | 16 |
| Cases with sequences | 102 | 25 | 7 | 2 | 136 |
| Complete coding genomes[e] | 36 | 16 | 0 | 2 | 54 |
| At least N-X/P genes[e] | 66 | 9 | 7 | 0 | 82 |
| **Publicly available sequences** | | | | | |
| Total cases included | 55 | 16 | 36 | 3 | 110 |
| Complete coding genomes[e] | 9 | 15 | 10 | 2 | 36 |
| At least N-X/P genes[e] | 46 | 1 | 26 | 1 | 74 |
| **Total cases included** | 286 | 47[f] | 43 | 5 | 381 |
| Confirmed BoDV-1 infections[g] | 262 | 46[f] | 43 | 5 | 356 |
| Cases with sequences | 157 | 41 | 43 | 5 | 246 |
| Coding-complete genomes | 45 | 31 | 10 | 4 | 90 |
| At least N-X/P genes | 112 | 10 | 33 | 1 | 156 |
| Total cases with available location | 278 | 43 | 42 | 2 | 365 |
| Confirmed with available location[g] | 254 | 42 | 42 | 2 | 340 |
| Confirmed with available year[g] | 262 | 46 | n.a. | 3 | 309 |
| Confirmed with available month[f] | 257 | 45 | n.a. | 0 | 302 |
| Sequences with available location | 155 | 39 | 42 | 2 | 238 |

[a]Domestic mammals also include two cases of non-domesticated zoo animals (pygmy hippopotamus).

[b]BoDV-1 isolates He/80, strain V, H24, H215 and DessauVac, which all originate from domestic mammals and have passaging histories in cell culture and/or experimental animals extending beyond the initial isolation in cell culture, were classified as laboratory strains. If more than one sequence per isolate was available, only the original sequence was included. The cell culture-derived materials from cases H640 and H3053 are not included in this table since they were used only for re-sequencing and correction of sequence database entries.

[c]Human samples analysed in this study originated from unpublished cases as well as from previously published cases without published BoDV-1 sequence[7,9,13,14,16–18,43,45].

[d]n.a. = not analysed.

[e]Complete coding BoDV-1 genomes: 8,769 nucleotides, ranging from genome position 54 (start of the N gene) to 8822 (end of the L gene); N-X/P sequences: 1,824 bp, ranging from position 54 to 1877 (end of the P gene).

[f]In addition to the 29 human cases analysed during this study and the 16 human cases with publicly available sequences, two further published human cases without available sequence were regarded as confirmed human BoDV-1 infections based on their unequivocal epidemiological link in combination with detectable seroconversion. These two patients are the donor and the surviving liver recipient of the solid organ transplant cluster published by Schlottau et al.[8].

[g]Cases confirmed by either positive RT-qPCR result or publicly available sequences.

described clusters but formed a separate cluster 5 basal to them (Figs. 2 and 3A and Supplementary Fig. 5).

To provide a more objectifiable basis for BoDV-1 cluster designation, we performed further analyses based on pairwise nt sequence identities of complete coding genomes (Supplementary Fig. 6). Sequences belonging to the same cluster shared at least 96.0% pairwise nt sequence identity, whereas sequences of different clusters were only up to 95.7% identical. Cluster 1 sequences shared 96.0 to 96.4% nt sequence identity between subclusters 1A and 1B and at least 96.9% within each of the two subclusters, thus providing objectifiable demarcation criteria for cluster and subcluster assignment of complete coding genome sequences (Supplementary Fig. 6). In agreement with these values, the two sequences of the novel cluster 5 possessed 99.0% nt sequence identity with each other, but only 93.8 to 95.1% nt sequence identity with any other BoDV-1 sequence, supporting their affiliation to a separate cluster.

**Temporal and spatial relationships and host-association within BoDV-1 clusters**

In agreement with the well-known genetic stability of orthobornaviruses[38,40,41], we observed pairs of nearly identical BoDV-1 sequences that were detected many years or even several decades apart from each other (Fig. 2 and Supplementary Fig. 5). This observation was further confirmed by linear regression analysis of the ML tree root-to-tip divergence of BoDV-1 sequences over the past 43 years, which demonstrated the year of sampling to have no significant effect on the genetic divergence of the phylogenetic clusters and subclusters ($R^2 = 0.0057$ to $0.0570$; $P > 0.05$; Supplementary Fig. 7).

Furthermore, the phylogenetic analysis did not reveal host-specific clades. With the exception of the novel cluster 5, all clusters and subclusters were composed of closely related sequences derived from shrews and domestic mammals. Human sequences were identified in all clusters and subclusters except for subcluster 1B (Fig. 2 and Supplementary Fig. 5).

In contrast, the isolation by distance (IBD) analysis of 238 BoDV-1 N-X/P gene sequences with available location suggested a significant positive correlation between geographic and genetic distance (Mantel test $p < 0.0001$; Supplementary Fig. 8A). This positive correlation was also found for the individual BoDV-1 clusters ($p \le 0.0028$; Supplementary Fig. 8A; summarised in Fig. 3A).

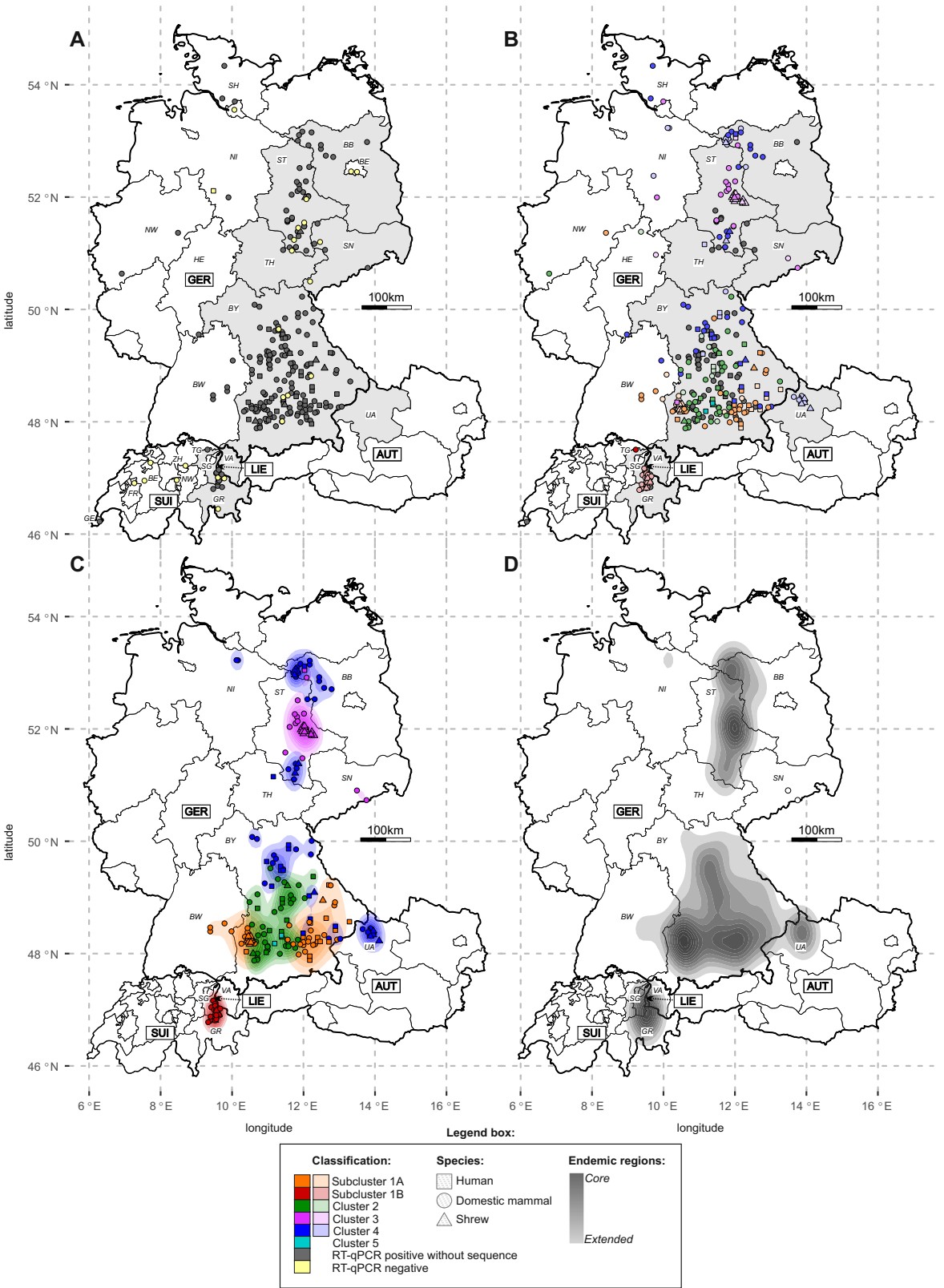

**Legend box:**

**Classification:**
- Subcluster 1A
- Subcluster 1B
- Cluster 2
- Cluster 3
- Cluster 4
- Cluster 5
- RT-qPCR positive without sequence
- RT-qPCR negative

**Species:**
- Human
- Domestic mammal
- Shrew

**Endemic regions:**
- Core
- Extended

## Detailed spatial analysis of BoDV-1 phylogenetic clusters and subclusters

For a detailed analysis of the spatial distribution of BoDV-1, subtrees of individual clusters or subclusters were extracted from the BoDV-1 N-X/P ML tree (Fig. 4A and Supplementary Fig. 5) and all cases with available location were mapped geographically (Fig. 4B−G). To aid visualisation of relationships between phylogenetic analysis and geographic mapping, we indicated monophyletic subclades showing associations to particular geographic regions (Fig. 4 and Supplementary Fig. 5).

The spatial distribution of subcluster 1A covers parts of south-eastern BY as well as an area from southwestern BY to eastern Baden-Wuerttemberg (BW; Fig. 4B and Supplementary Fig. 5A). This bipartite distribution is also reflected by the phylogenetic pattern. The sequences from southeastern BY are located on a common branch that

**Fig. 1 | Geographic location of analysed cases and BoDV-1 sequences. A** Origin of suspected or confirmed cases of Borna disease from domestic mammals and humans and of BoDV-1-infected shrews submitted for analysis. Grey symbols represent cases confirmed by BoDV-1-specific RT-qPCR in this study. Yellow symbols represent cases without a positive RT-qPCR result. The federal states (Germany, Austria) and cantons (Switzerland) coloured in light grey represent the assumed endemic regions based on previously published work[4,7,25,33,38,55]. **B** Geographic locations of BoDV-1 sequences originating from this study (dark colours) or previously published cases (light colours). Colours represent phylogenetic BoDV-1 clusters and subclusters as determined in Figs. 2 and 3A and Supplementary Fig. 5. Grey symbols represent cases confirmed by BoDV-1-specific RT-qPCR in this study without available sequence. **C** Visualisation of endemic regions of BoDV-1 clusters and subclusters by Kernel Density Estimation (KDE). The analysis is based on 214 BoDV-1 sequences with available location. Sequences classified as phylogenetic outliers (no additional sequence with at least 98.6% nucleotide sequence identity within a maximal distance of 37.9 km) were excluded from the analysis. **D** Cluster-independent BoDV-1 endemic region visualised by KDE. Only sequences meeting the criteria described for panel C) were included. Germany (GER): BB Brandenburg, BE Berlin, BY Bavaria, BW Baden-Wuerttemberg, HE Hesse, NI Lower Saxony, NW North Rhine-Westphalia, SH Schleswig-Holstein, SN Saxony, ST Saxony-Anhalt, TH Thuringia; Switzerland (SUI): BE Bern, FR Fribourg, GE Geneva, GR Grisons, NW Nidwalden, SG St. Gall, TG Thurgau, ZH Zurich; Austria (AUT): UA Upper Austria, VA Vorarlberg; Liechtenstein (LIE).

is subdivided into three phylogenetically and spatially distinguishable subclades (1A.SE-1, -2 and -3) covering distinct regions in southeastern BY with only minor overlaps among each other. The sequences from southwestern BY form a separate subclade (1A.SW). A further subclade is constituted by four sequences from BW (1A.BW-1), while the phylogenetic position of the fifth sequence from BW (1A.BW-2) is separated from all other sequences (Fig. 4A and Supplementary Fig. 5A).

Subcluster 1B is restricted to a rather confined region in the Alpine Rhine valley in the Swiss cantons Grisons (GR) and St. Gall (SG), with a few cases detected across the border into Liechtenstein (Fig. 4C). This subcluster can be subdivided into a southern and a northern subclade (1B.S and 1B.N, respectively; Fig. 4C and Supplementary Fig. 5B). In this study, we were able to generate only one new sequence of subcluster 1B, which originated from the canton Thurgau (TG), a neighbouring canton of SG that has not been described as endemic for BoDV-1 so far. In agreement with its separate location, the phylogenetic position of this sequence is basal to subclades 1B.S and 1B.N (Fig. 4C and Supplementary Fig. 5B).

Cluster 2 occupies major parts of central BY, mainly covering the region between the two parts of the dispersal area of subcluster 1A (Figs. 1B and 4D; and Supplementary Fig. 5C). The cluster can be subdivided into three monophyletic subclades in the southwestern part of this area (2.SW-1, -2 and -3) and one that is mainly found more to the northeast in central BY (2.MID) with only a few overlapping cases (Fig. 4D and Supplementary Fig. 5C).

Cluster 3 is situated mainly in eastern ST and in the South of SN, with single outliers in other federal states (Fig. 4E and Supplementary Fig. 5D). We assigned two phylogenetically supported subclades with, however, strongly overlapping dispersal areas (3.GG and 3.RO, named after the locations of the majority of their mainly shrew-derived sequences, Güterglück and Rosslau, respectively). The remaining sequences could not be assigned to prominent phylogenetic subclades (Fig. 3E and Supplementary Fig. 5D).

Cluster 4 is the most widely distributed cluster with a scattered geographic range covering parts of Upper Austria (UA), northern and southeastern BY, ST and BB. Individual additional cases were located in Schleswig-Holstein (SH), NI, Thuringia (TH), Hesse (HE) and BW (Fig. 4F and Supplementary Fig. 5E). In agreement with this geographic pattern, sequences of cluster 4 form several subclades with apparent association with distinct regions in UA (4.UA), southeastern and northern BY (4.BY-SE, 4.BY-N-1 and -2) and BB (4.BB). No clear subclades could be defined for a large group of genetically rather diverse sequences from northern parts of Germany (Fig. 4F and Supplementary Fig. 5E).

Both sequences of the newly identified cluster 5 originated from two neighbouring districts in a region in southern BY where usually sequences of cluster 2, subclades 2.SW-1 and -2, were found (Figs. 1B and 4D, G and Supplementary Fig. 5F).

## Definition of phylogeographic outliers
While phylogenetic grouping and geographic mapping were in good agreement with each other for the majority of BoDV-1 sequences,

singular sequences appeared to be located distant from their phylogenetic relatives (Fig. 4). We sought for objective and reproducible criteria to define such sequences as phylogeographic outliers. Pairwise nt comparisons of all N-X/P-gene sequences with available location ($n = 238$) showed that all sequences possessed one or more relatives with at least 98.6% nt sequence identity, with only one exception sharing only 98.1% nt sequence identity to its next relative (Supplementary Fig. 9A; representing sequence OR468948, marked as case F in Fig. 4C, Supplementary Fig. 5B and Supplementary Table 3). The minimum distances of each case to all sequences with at least 98.6% nt sequence identity ranged from 0 to 366 km, with the 90th percentile at 37.9 km (Supplementary Fig. 9B). Based on these observations, all cases without a sequence of at least 98.6% nt sequence identity within a maximum distance of 37.9 km were marked as outliers, corresponding to the approximately 10% sequences with the highest minimal spatial distance to any close relative. These criteria were met by 24 of the 238 N-X/P sequences (10.1%; Fig. 4, Supplementary Fig. 5 and Supplementary Table 3). The same 24 cases were identified as phylogeographic outliers by a comparative approach using patristic distances instead of nt sequence identities (Supplementary Table 10). Removal of outlier sequences from the IBD analysis increased the correlation coefficient (*r*), which measures the strength and direction of the correlation between genetic and geographic distance, for all clusters with the exception of cluster 1B. This effect was most prominent for cluster 2 (Supplementary Fig. 8B; summarised in Fig. 3B).

## Detailed characteristics of defined phylogeographic outliers
For four of these outliers (outliers A, B, K and L), the available records revealed potential epidemiological links to areas where the respective BoDV-1 variants were considered to be endemic (Supplementary Table 3). A horse (outlier A; accession OR468845) had developed Borna disease in 2019 in North Rhine-Westphalia (NW), which is not known to be endemic for BoDV-1. The animal had been purchased from an endemic region in southwestern BY approximately two months before its death. In line with this information, the BoDV-1 N-X/P sequence from this horse belonged to subclade 1A.SW and it was identical to a sequence from a sheep from southwestern BY in 2009 (OR468934; Fig. 4B, Supplementary Fig. 5A and Supplementary Table 3). An alpaca stallion from northern BY in 2022 (outlier B; OR468886) had been bought eight months before death from a region in southeastern BY, which is consistent with its BoDV-1 sequence belonging to subclade 1A.SE-2 (Fig. 4B, Supplementary Fig. 5A and Supplementary Table 3). The animal showed ataxia already on arrival at the new herd, which was assumed to be of orthopaedic cause. An additional alpaca stallion (outlier K; GQ861449) had developed disease in 2008 after having been transported from southwestern BY to the north of HE [21,21]. The BoDV-1 sequence from this case belonged to subclade 2.SW-1, which is in congruence with an infection source in south-western BY (Fig. 4C Supplementary Fig. 5B and Supplementary Table 3). Outlier L (OR468852), a horse from northern BY in 2021, had been bought from a horse trader in BW two weeks before death. The horse was reported to have been bought by the trader several weeks

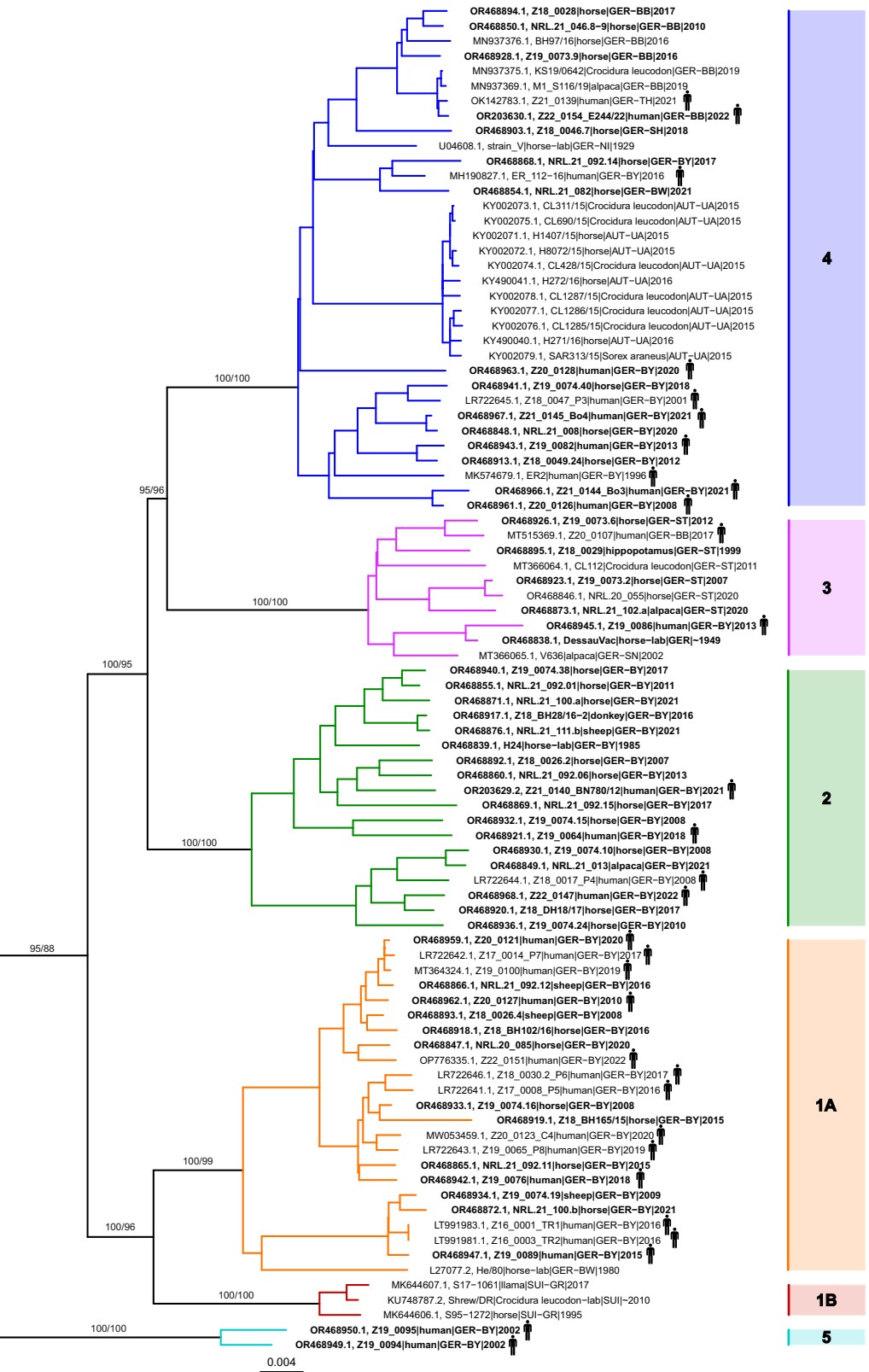

**Fig. 2 | Phylogenetic analysis of complete coding BoDV-1 genome sequences.** A maximum likelihood tree (model SYM + G4) was calculated for all 90 complete coding BoDV-1 sequences (genome positions 54 to 8822) of human and animal origin. Sequence BoDV-2 No/98 (AJ311524; not shown) was used to root the tree. Sequences generated during this study are depicted in bold. Statistical support is shown for major branches, using the format "SH-aLRT/ultrafast bootstrap".

Clusters 2 to 5 and subclusters 1A and 1B are indicated by coloured branches and bars. Germany (GER): BB Brandenburg, BY Bavaria, BW Baden-Wuerttemberg, NI Lower Saxony, SH Schleswig-Holstein, SN Saxony, ST Saxony-Anhalt; Switzerland (SUI): GR Grisons; Austria (AUT): UA Upper Austria. Phylogenetic tree in Newick format is provided in Supplementary Data 2.

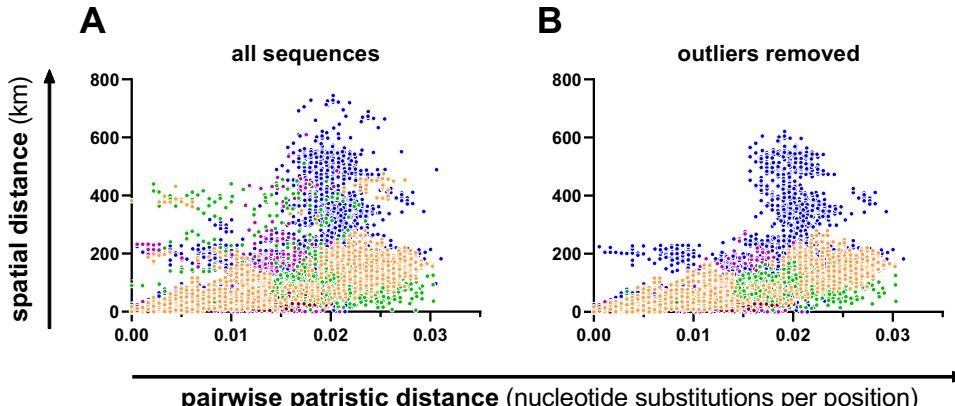

**Fig. 3 | Illustration of pairwise spatial versus phylogenetic distances.** Pairwise patristic distances were inferred from the maximum likelihood (ML) tree of N-X/P nucleotide sequences (Fig. 4A and Supplementary Fig. 5) and plotted against pairwise spatial distances. **A** Results of all 238 N-X/P nucleotide sequences with available locations. **B** Results of 214 N-X/P sequences after removal of the 24 sequences that were categorised as phylogeographic outliers using the following criteria: existence of no other BoDV-1 N-X/P sequence with ≥98.6% nucleotide sequence identity within a distance of ≤37.9 km. Colours represent the phylogenetic clusters 2 to 5 and subclusters 1A and 1B (see Fig. 2). Only pairwise comparisons within clusters or subclusters are depicted in this figure. A detailed presentation of the analysis is shown in Supplementary Fig. 8.

before from a not further specified location in BY. The BoDV-1 sequence of subclade 2.SW-1 suggests an infection source in southwestern BY (Fig. 4C; Supplementary Fig. 5B; Supplementary Table 3).

In contrast, no epidemiological link to a potentially aberrant location of infection could be identified for the majority of the 24 identified outliers. In some cases, the available information suggested that the animal had never been in a region considered endemic for the respective BoDV-1 variant (outliers J, P, U), but in most cases the available information was insufficient. In some cases, the accuracy of the location data was low, e.g. representing the owner´s address, which may be distant from the actual husbandry (Supplementary Table 3). The human case Z19_0093 from western BY in 2016 (outlier F; OR468948) was classified as an outlier, since its sequence possessed only up to 98.1% nt sequence identity to any other BoDV-1 sequence.

Two further RT-qPCR-confirmed BoDV-1 infections in horses were detected far from any known BoDV-1-endemic region. These cases had occurred in western Switzerland (canton Geneva [GE]) in 1988 and in eastern BB in 2006 (Fig. 1B). However, sequencing attempts had failed due to poor RNA quality. Epidemiological data were not available for these cases.

### Phylogeographic relationship of human- and animal-derived BoDV-1 sequences
Of the 43 non-transplant-derived human BoDV-1 infections confirmed prior to preparation of this manuscript, 40 were diagnosed in BY, two in BB and one in TH (Figs. 1B and 4). The vast majority of their BoDV-1 sequences matched the phylogenetic subclades found in the respective patient´s region of residence (Fig. 4). Typically, sequences originating from animals or humans in close geographic proximity were representing the genetically closest relatives of human-derived sequences. For 90% of human-derived N-X/P sequences with available location ($n = 39$), the distance to their phylogenetically closest relatives was less than 40 km, with a median distance of 15.6 km (Fig. 5). In several cases, identical animal-derived sequences or sequences sharing ≥99.9% nt sequence identity were found within <10 km distance to the residency of the respective human patient. For instance, the human BoDV-1 sequence Z21_0129 (OR468964; subclade 1A.SW) shared 99.9% nt sequence identity with the horse-derived sequence OR468847 (NRL.20_085) that originated from the same district in south-eastern BY (Supplementary Fig. 5A). Similarly, human sequence Z19_0107 (MT515369; cluster 3)[12] was 99.9% identical to that of an alpaca (NRL.22_102; OR468885) from the same district in BB

(Supplementary Fig. 5C). The BoDV-1 N-X/P sequences from patients Z19_0100 (MT364324)[11,15] and Z20_0121 (OR468959) were completely identical among each other (99.9% at complete genome level) and 99.9% identical to the sequence of sheep NRL.21_092.12 (OR468866) from the same district in southeastern BY (Supplementary Fig. 5A).

Besides outlier F mentioned above, only one additional human sequence was classified as an outlier. The BoDV-1 sequence of patient Z19_0086 (OR468945; outlier N; Supplementary Table 3) from southwestern BY belonged to cluster 3 and was, thus, genetically divergent from all other sequences originating from this area. Its sequence possessed 99.9% nt sequence identity to the nt sequence of the vaccine strain 'DessauVac' (Supplementary Fig. 5C). The precise origin of this historic vaccine strain is unknown, but it is believed to have been isolated from a horse from ST or western SN around 1949[30,38]. The closest relative of sequence Z19_0086 with available location originated from an alpaca in SN (MT366065), about 370 km from the patient´s residency (Supplementary Table 3 and Fig. 5). Information on possible epidemiological links to ST or SN was not available for patient Z19_0086. Likewise, no information was available on potential contacts to the vaccine strain DessauVac, which had been used until 1992 in eastern parts of Germany (former German Democratic Republic), but never in BY. The sequence of case Z21_0139 from TH, 2021 (OK142783; marked as case #1)[13], showed a high spatial distance to its most closely related BoDV-1 sequences. Its sequence belonged to subclade 4.BB and possessed up to 99.9% nt sequence identity to animal sequences from BB that are located more than 200 km from the patient´s home (Fig. 5), suggesting an aberrant infection site. However, since less closely related additional cluster 4 sequences (99.3% nt sequence identity) were found at a distance of roughly 25 km, the case did not match our criteria for a phylogeographic outlier (Supplementary Table 4).

Interestingly, both patients infected with BoDV-1 of cluster 5 had died after developing disease in 2002 and both lived in neighbouring districts in the vicinity of Munich (BY; Fig. 4G and Supplementary Fig. 5F). Cluster 5 has not been detected in shrews or domestic mammals so far.

### Determination of BoDV-1 endemic areas
To visualise the endemic areas of each BoDV-1 cluster or subcluster, we employed Kernel Density Estimation (KDE; Fig. 1C). For this approach, all phylogeographic outliers defined above (Supplementary Table 3) were excluded from the dataset. The resulting KDE illustrated that the

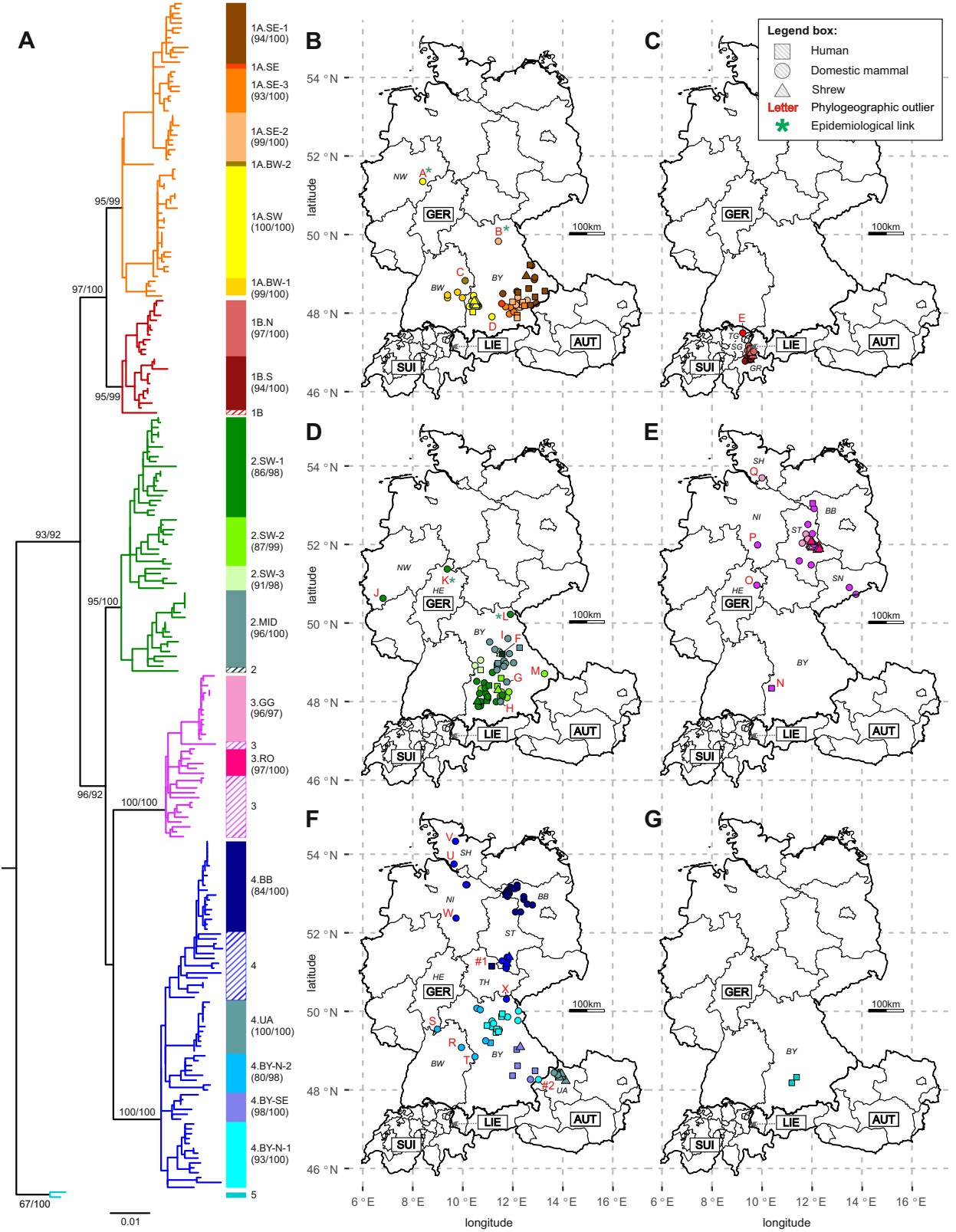

distribution of each BoDV-1 cluster or subcluster covered largely separate areas with little overlap (Fig. 1C). In a second approach, we repeated the KDE using the combined dataset of all BoDV-1 clusters. This combined KDE revealed a tripartite endemic area (Fig. 1D). The northernmost part ranges from northwestern BB to southern ST, possibly also including northern TH. The largest part covers most of BY and extends into BW and UA. The southernmost endemic area is found in the Alpine Rhine valley (Fig. 1D). In addition, individual sequences not classified as outliers are found in NI south of Hamburg and in the Ore Mountains in SN, close to the Czech border (Fig. 1D).

## Discussion

The aim of this study was to assemble the most comprehensive data on the molecular epidemiology and phylogeography of BoDV-1 allowing

**Fig. 4 | Detailed phylogeographic analysis of BoDV-1 clusters and subclusters.**
**A** A maximum likelihood (model GTR + F + I + G4) tree was calculated for 246 partial BoDV-1 sequences (1,824 nucleotides, nt) of human and animal origin that are covering the complete N, X and P genes (genome positions 54 to 1877). Sequence BoDV-2 No/98 (AJ311524; not displayed) was used to root the tree. Statistical support is shown for main branches (including clusters, subclusters, and subclades), using the format "SH-aLRT/ultrafast bootstrap". Clusters 2 to 5 and subclusters 1A and 1B are indicated by coloured branches. Subclades are indicated by coloured bars and corresponding text labels, with statistical support of subclades shown in brackets. **B–G** Spatial distribution of subclusters 1 A (**B**) and 1B (**C**) and clusters 2 (**D**), 3 (**E**), 4 (**F**) and 5 (**G**). Colours of the symbols represent the phylogenetic

subclades indicated in panel A). Human sequences are generally mapped no more precise than to the centre of the district of the patient´s residence. Red labels represent phylogeographic outliers (capital letters; Supplementary Table 3) or additional cases with potentially aberrant infection site (#1 and #2; Supplementary Table 4). Green asterisks indicate known epidemiologic links into the dispersal area of the respective subclade. Germany (GER): BB Brandenburg, BY Bavaria, BW Baden-Wuerttemberg, HE Hesse, NI Lower Saxony, NW North Rhine-Westphalia, SH Schleswig-Holstein, SN Saxony, ST Saxony-Anhalt, TH Thuringia; Switzerland (SUI): GR Grisons, SG St. Gall, TG Thurgau; Austria (AUT): UA Upper Austria; Liechtenstein (LIE). Subclade designations: GG Güterglück, MID Middle, N North, S South, RO Rosslau, SE Southeast, SW Southwest.

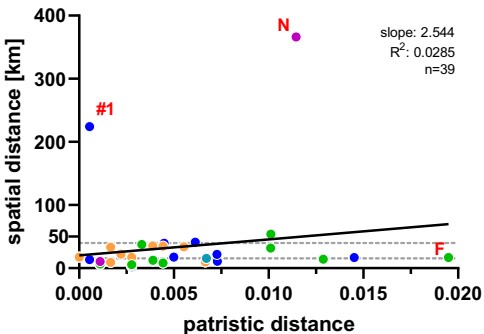

**Fig. 5 | Geographic distance of human BoDV-1 sequences to their closest phylogenetic relatives.** The minimal distance to the most closely related BoDV-1 nucleotide (nt) sequence was identified for all human BoDV-1 sequences with available geographic information ($n = 39$) based on patristic distances calculated from the maximum likelihood (ML) tree of 246 N-X/P nt sequences (Supplementary Fig. 5). Sequences without available location as well as non-human sequences classified as phylogeographic outliers (Supplementary Table 3) were excluded from the analysis. For each human sequence, the minimal spatial distance was calculated to all sequences with a patristic distance of up to 1.2-fold the patristic distance to the phylogenetically closest relative. Colours of the dots represent phylogenetic clusters and subclusters as defined in Fig. 2. Red capital letters indicate human cases identified as phylogeographic outliers (Supplementary Table 3). Sequence Z21_0139 (labelled as #1) is suspected to have an aberrant infection site due to its close genetic relation to animal sequences in more than 200 km distance (Supplementary Table 4). Broken horizontal lines represent the 90th percentile (39.8 km) and the median (15.6 km) of the presented dataset. The black line represents the linear regression of genetic and geographic distance. Slope and goodness of fit ($R^2$) of the regression line are provided.

the identification of risk areas for the occurrence of spill-over transmission to domestic mammals and humans. Detection of BoDV-1 in shrews as the only known reservoir hosts would provide the most accurate information on its endemic presence. However, representative samples from shrews are not easily accessible. To date, these data are highly fragmented and biased due to active sampling at only a few locations[4,25,31–34,42]. We therefore adopted a passive surveillance approach, utilising Borna disease in domestic mammals as an indicator of endemic BoDV-1 infection in local shrew populations. This approach may provide a potentially less biased fundament for phylogeographic analyses, although some variability of the dissemination of susceptible domestic animal populations and of veterinary vigilance within and outside of known endemic areas cannot be excluded. Reliable information on the location of infected individuals, not only during onset of disease but even more importantly, at the potential time point of infection, is crucial for this study. However, due to the long and possibly highly variable incubation period, such information is not always available, particularly for cases from earlier decades. Despite our extensive efforts to fill these data gaps, there are still varying degrees of uncertainty that must be considered when interpreting the results of this study.

Our analyses supported the previously introduced phylogenetic clusters and subclusters[38–40], for which we established objectifiable demarcation criteria, based on pairwise nt sequence identities of complete coding BoDV-1 genomes. The identification of a novel BoDV-1 cluster 5, represented by two human sequences from BY, indicates that the genetic variability of BoDV-1 may be higher than currently appreciated and that additional variants may exist within or outside the known endemic areas.

We found the BoDV-1 clusters and subclusters to be genetically remarkably stable over time and to be associated with spatial distribution rather than time of detection or host species, as demonstrated previously[7,25,38,40]. In our study, we were able to markedly increase the extent, reliability and resolution of the phylogeographic data, showing that BoDV-1 sequences of particular subclades are occupying circumscribed areas within the endemic regions. This geographically bound epidemiology further emphasises that BoDV-1 is tied to a reservoir with a strictly territorial behaviour and only very little mobility[38,40]. Bicoloured white-toothed shrews are known to occupy territories of 40 to 120 metres in diameter. They typically do not move voluntarily more than 800 m to 1 km from their territory, with a documented maximum of 2.5 km, thus allowing for only limited virus spread[36]. This spatial limitation also underlines the fact that spill-over hosts, such as domestic mammals and humans, which tend to be more mobile, serve as dead-end hosts for the virus. Otherwise, their contribution to virus spread would have resulted in a much wider distribution of confirmed cases with considerable spatial overlap of genetic variants. Thus, our results clearly support previous work[35,39,40,46] and refute former hypotheses of a possible worldwide spread of BoDV-1 among humans and non-reservoir animals[47,48].

To indicate the regions in which BoDV-1 variants of particular clusters or subclusters are endemic, we defined criteria for those sequences that indicate the endemic presence of the virus with reasonable reliability. Due to the limitations described above, a single BoDV-1 detection in a domestic mammal or human spatially separated from its closest viral relative cannot be considered as an indicator of endemicity. It rather needs to be supported by additional genetically related sequences from the same region. We have tentatively defined the criteria for considering sequences as indicators of endemicity to be at least two sequences with at least 98.6% nt sequence identity within a 37.9 km diameter. Applying these criteria, the ~10% of the sequences with the highest spatial distance from their closest relatives are regarded as phylogeographic outliers and excluded from further analysis. We used N-X/P nt sequence identities rather than patristic distances to make the analysis more easily applicable to further cases diagnosed in the future. A comparative analysis confirmed that using patristic distances led to consistent results, demonstrating the robustness of our approach. The criteria defined in this study may be subject to further refinement as more extensive data will become available. Visualisation of the distribution of the BoDV-1 sequences after removal of the phylogeographic outliers confirmed and refined the previously assumed pattern of endemic areas[35], but also extended it in certain regions, such as northern BY and eastern BW.

Geographic outliers do occasionally occur distant from the indicated distribution areas of their phylogenetic clades or even completely outside the defined endemic area of BoDV-1. Due to the limitations of the available metadata, it cannot be excluded that at least some of these cases may be the result of inaccurately reported locations. Others may be the result of travel within the incubation period, as previously described[21,22]. During the course of this study, at least three additional cases were identified in which animals were likely to have been moved to new locations during the incubation period, with BoDV-1 sequences suggesting sources of infection at the respective sites of origin. In one of these cases, the alpaca had been moved no less than eight months prior to death. However, the incubation period may have been shorter, as the animal was exhibiting a potentially BoDV-1-associated ataxia already at the time of transfer.

Such clear epidemiological links could not be established for the majority of phylogeographic outliers identified in this study. While no further information was available for most of these cases, some animals were reported to have never travelled to known endemic areas, or even to have remained in their holding of birth throughout their lives, thus suggesting the existence of so far unidentified infection sources in these regions. Whether or not BoDV-1 can be transmitted over long distances by passive vectors, such as import of contaminated feed, remains elusive. So far, no such cases have been documented and data on the environmental tenacity of the virus are sparse. Thus, at least some of the latter cases may actually indicate the endemic presence of the virus in local shrew populations, requiring confirmation by detection of additional genetically related viruses from their region. For instance, the phylogenetic analyses of the previously published human case Z19_0107 from BB had grouped the virus to cluster 3 that had not been detected in this region before[12], suggesting a possibly aberrant infection source. The recent detection of a highly similar BoDV-1 sequence from an alpaca from the same district in BB now rather suggests endemicity of this BoDV-1 variant in this region and a peridomestic infection of both individuals, which is also in accordance with the patient´s epidemiological history during the last years before onset of the disease[12].

Our study increases the number of published laboratory-confirmed human BoDV-1-infections to 46 and provides a first comprehensive summary of metadata for all published cases[6-18]. The cases covered all age groups and encephalitis was diagnosed in 45 of the 46 patients, 44 of whom had died as a result of the disease, resulting in a known case-fatality rate of 97.8%. As of now, the representativeness of these numbers for all incident cases is difficult to assess. Many cases in this study were diagnosed only by retrospective analysis of encephalitis cases. Given the only recently discovered human threat of the virus, it is likely that particularly in the past a considerable proportion of fatal BoDV-1 infections remained undetected[7]. This proportion may have been comparably higher for possible non-fatal infections due to a lower alertness of physicians to such cases and to the limited availability of diagnostic material for direct intra vitam detection of BoDV-1, which is hampered by the remarkable restriction of BoDV-1 to the central nervous system in erroneous spill-over hosts[1,7,9,20,25,35,49]. However, the estimated prevalence of bornavirus-reactive antibodies did not exceed 0.24% in recent serological surveys of healthy individuals or neuropsychiatric patients in known endemic areas[43,45,50]. Analysis of brain tissue from encephalitis cases of unknown origin failed to detect BoDV-1 in biopsy samples from 15 non-fatal cases, while BoDV-1 was detected in seven out of nine (78%) fatal cases from the same cohort[7]. Likewise, nation-wide screenings identified bornavirus-reactive antibodies in serological samples and/or BoDV-1 RNA in CSF only in patients suffering from severe or fatal encephalitis[11,43]. These findings support the assumption that non-fatal or even asymptomatic human BoDV-1 infections are indeed at least very rare.

The detailed phylogeographic network of animal-derived BoDV-1 sequences assembled in this study also allowed for a first phylogenetic assessment of potential geographic sources of human BoDV-1 infections. With the exception of three organ transplant-derived infections[8], all known infections are assumed to have resulted from individual zoonotic spill-over events from the virus reservoir[6,7,9-12], even though concrete transmission events could not be identified in a retrospective epidemiological analysis of 20 cases[17]. Almost all human BoDV-1 sequences clustered in accordance with the location of the patient´s residence. The median estimated distance to the closest phylogenetic relatives was 15.6 km, indicating that infection of most patients occurred close to home, which is in accordance with epidemiological work identifying rural residence on the fringe of the settlement as the major risk factor for BoDV-1 infection[17]. The actual distance to the source of infection is likely to be even lower in many cases, since inaccuracies of location data, as well as unavailability of further sequences representing genetically closer relatives are likely to lead to overestimation rather than underestimation.

The exact source and route of transmission remain elusive for almost all BoDV-1 infections in humans and domestic mammals. Due to its almost exclusively neurotropic nature in non-reservoir hosts[7,9,20,25,49], transmission chains between spill-over hosts can be virtually excluded, with the exception of a so far singular iatrogenic transmission event by solid organ transplantation insert ref. 8. This leaves infected shrews as the most likely infection source. So far, it remains unknown whether exposure to their excretions is sufficient for transmission or whether direct contact to an infected shrew or its carcass is required. In a previous study, household members of deceased patients could not recollect potential events of BoDV-1 exposure, indicating that these may be rather unremarkable[17]. Experimentally, BoDV-1 infection in animal models may be established via mucosal surfaces, mainly intranasally, or by subcutaneous injection, followed by axonal spread to the brain via nerves and olfactory bulb or spinal cord[23,51-54]. Overall, spill-over transmission of BoDV-1 to humans appears to be rather inefficient, as only isolated cases have been reported so far. Associated infections of potentially equally exposed individuals, such as family members or co-workers on agricultural farms, have not been detected, yet. While the disease usually affects only a single animal or a small number of individuals in herds of horses and sheep, higher incidences of BoDV-1 infections have been observed during outbreaks in New World camelid holdings, leading to mortality rates of up to 40% within a few months in affected herds[25,38,40,55]. Similarly, four out of eight alpacas of a herd from central BY analysed in our study succumbed to confirmed BoDV-1 infection within one year (cases 21_013, 21_149.a, 22_015.a and 22_015.b; Supplementary Fig. 5C).

Previous studies have hypothesised a higher risk of BoDV-1 transmission to domestic mammals during winter, leading to disease outbreaks and eventually death during spring and early summer. Shrews entering animal stables in search of feed have been suggested to be responsible for this pattern[33,40]. In congruence with these assumptions, our assembled data suggested a peak of laboratory-confirmed fatal Borna disease in domestic mammals during May and June. In contrast, no such seasonality was observed for the time of death or first known hospitalisation of the comparably limited number of human BoDV-1 cases, similar to findings in a previous study of 20 human cases[17]. However, any conclusions regarding the time of infection are complicated by the unknown incubation period and the variable disease progression. Furthermore, the time from infection to death of human patients is affected by attempted treatments and life-sustaining measures, which further increase the variability. A larger dataset of human BoDV-1 infections may be required to demonstrate whether their temporal distribution actually differs from the occurrence of BoDV-1 infection in domestic mammals.

In summary, we performed a highly comprehensive phylogeographic analysis of the occurrence of the zoonotic pathogen BoDV-1 in Central Europe. The improved resolution of the phylogeographic data

will provide a basis for assessing potential locations and sources of BoDV-1 infection in animals and humans. The visualisation of potential risk areas will allow for the implementation of prophylactic measures – mainly reducing the risk of exposure to the reservoir – in the affected regions. In such areas, BoDV-1 may be responsible for a considerable proportion of cases of severe human encephalitis that may have remained unresolved so far[7]. Increased awareness among veterinarians and physicians, together with the categorisation of BoDV-1 as a notifiable pathogen of humans and animals since 2020[11], may lead to more extensive data collection, allowing for further refinement of phylogeographic analyses in the future.

## Methods

### Acquisition of sample material

Veterinary pathologists and federal and private veterinary diagnostic laboratories in Germany, Switzerland and Austria were informed about the study through presentations at scientific meetings, publications in national specialist journals, via mailing lists of expert societies and by direct contact. In total, 20 institutions provided fresh-frozen or FFPE brain tissue or CSF from 231 archived or recent suspected BoDV-1 infections in domestic mammals (including few zoo animals; Table 1; Supplementary Table 1). Some, but not all, of these infections had already been diagnosed by the submitting diagnostic laboratories. In addition, brain samples from 29 archived or recent human BoDV-1 encephalitis cases were obtained from diagnostic centres and pathologists in Germany. These cases included unpublished cases as well as previously published cases without available BoDV-1 sequence[7,9,14,16–18] (Table 1). In addition, samples from seven BoDV-1-positive bicoloured white-toothed shrews were obtained from an ongoing large-scale small mammal screening study (Haring et al., manuscript in preparation; Table 1). Furthermore, an original vaccine vial containing the historic BoDV-1 live vaccine strain 'Dessau', herein referred to as 'DessauVac' (batch 193 02 90; kindly provided by Sven Springer, IDT Biologika, now Ceva Santé Animale, Dessau-Rosslau, Germany), and the cell culture isolate H24[39] were included for sequence analysis. In addition, the horse-derived cell culture isolates H640 and H3053[38] were kindly provided for resequencing by Sybille Herzog (Gießen, Germany).

### Acquisition of sample metadata

In order to facilitate spatio-temporal analyses, detailed metadata were requested from the submitters, including geographic location (postal code) and date of sampling, which was usually the date of death. Age, sex and date of hospital admission were recorded additionally for human cases. For animal cases, the accuracy of the geographic location was non-hierarchically categorised as follows: (1) the location of the animal husbandry is known, (2) the address of the owner is known, but the location of the husbandry is unknown and may be different, (3) only the submitting veterinary practice/clinic is known, (4) only the administrative district of origin is known, and (5) no information on the accuracy of the location is available.

In addition to the samples analysed in this study, previously published BoDV-1 sequences available via GenBank were included in the phylogeographic analysis (Table 1). The metadata described above (location, accuracy of location, host species, date of death or sampling) were assembled also for these cases, based on the available literature[4,6,7,9–13,15,21,25,33,38–40,55]. Missing data were completed by contacting the authors and/or the initial submitters, if possible.

The species of the seven analysed bicoloured white-toothed shrews had been confirmed by sequence analysis of the cytochrome *B* gene[56].

### Extraction of total RNA

Fresh-frozen samples were mechanically disrupted in 1 ml TRIzol reagent (Life Technologies, Darmstadt, Germany) by using the TissueLyser II (Qiagen, Hilden, Germany), according to the manufacturers' instructions. After the addition of 200 µl chloroform and a centrifugation step (14,000×g, 10 min, 4 °C), the aqueous phase was collected and added to 250 µl isopropanol. Total RNA was extracted using the silica bead-based NucleoMagVet kit (Macherey & Nagel, Düren, Germany) with the KingFisher™ Flex Purification System (Thermo Fisher Scientific, Waltham, MA, USA) according to the manufacturers' instructions.

Additional RNA extraction from fresh-frozen samples selected for HTS was performed according to Wylezich et al.[57]. Briefly, the tissue was rapidly frozen in liquid nitrogen and subsequently pulverised using the Covaris cryoPREP (Covaris, Brighton, UK). The resulting powdered tissue was then dissolved in pre-warmed 1 ml lysis buffer AL (Qiagen). RNA was extracted using the RNAdvance tissue kit (Beckman Coulter, Germany), including a DNase I (Qiagen) digestion step, in combination with a KingFisher Flex purification system (Thermo Fisher Scientific, Germany), according to the manufacturers' instructions. Total RNA was eluted in 100 µl nuclease-free water.

Total RNA from FFPE brain tissues was extracted as described previously[58]. Briefly, two FFPE sections of <10 µm thickness underwent deparaffinisation and proteinase K digestion employing the Covaris truXTRAC FFPE total NA kit before RNA extraction, according to the manufacturer's instructions, resulting in 100 µl supernatant. To prevent the transfer of paraffin residues, formalin de-crosslinking was carried out using 85 µl of the supernatant in a clean 1.5 ml reaction tube (80 °C, 30 min, thermomixer). Subsequently, 175 µl of B1 Buffer from the Covaris kit and 250 µl of 65% isopropanol were added, mixed, and briefly centrifuged. Subsequently, RNA extraction was performed using the Agencourt RNAdvance Tissue Kit, as described above.

### Detection of BoDV-1 RNA by RT-qPCR

BoDV-1 RNA was detected using two BoDV-1-specific RT-qPCR assays (Mix-1 & Mix-6) detecting P and M gene RNA, respectively (Supplementary Table 2), as described in detail elsewhere[8]. Exogenously supplemented, in vitro-transcribed RNA of the enhanced green fluorescence protein (eGFP) gene or host-derived beta-actin RNA were amplified as extraction control or RNA quality control, respectively, following previously described protocols[59,60] (Supplementary Table 2).

### Selection of samples for BoDV-1 sequencing

HTS was performed for 114 selected BoDV-1 cases, including the cell culture isolates H24 and DessauVac (Table 1). Selection criteria for animal samples included spatial proximity to known human BoDV-1 cases or the occurrence in regions from which no or only few BoDV-1 sequences were available. If more than one case was available from a particular location, samples with higher predicted ratios of BoDV-1 RNA (indicated by lower RT-qPCR Cq values) versus total RNA concentration were selected[61]. Enrichment of BoDV-1 specific library DNA fragments by hybridisation-based capture technology was performed for 16 selected samples for which insufficient BoDV-1 sequence information had been obtained by standard HTS (Table 1).

For additional 43 BoDV-1-positive fresh-frozen brain samples from domestic mammals, humans or shrews, partial BoDV-1 genome sequences were generated by Sanger sequencing (Table 1).

### High-throughput sequencing

Libraries with an average DNA fragment size of 500 bp were prepared from fresh-frozen BoDV-1-positive brain samples with sufficient RNA quality following the procedure described by Wylezich et al.[57] with modifications by Szillat et al.[62]. Modified library preparation protocols were used for FFPE-samples as well as for fresh-frozen samples with lower RNA quality, resulting in a mean DNA fragment size of 200 bp[7]. Library quantification was carried out with the QIAseq Library Quant Assay Kit (Qiagen) and fragment size of each library was analysed using Agilent High Sensitivity DNA kit implemented in 2100 BioAnalyzer Instrument (Agilent). Libraries of 500 bp fragment size were sequenced 400 bp runs using an Ion Torrent S5 XL instrument (Thermo

Fisher Scientific). Libraries of 500 bp DNA fragment size were sequenced in 400 bp runs using Ion 530 chips, while libraries of 200 bp DNA fragment size were sequenced in 200 bp runs using Ion 540 or Ion 550 chips on an Ion S5 XL instrument.

### BoDV-1 target enrichment by hybridisation-based capture technology

For enrichment of BoDV-1 specific library DNA fragments, an RNA bait set was designed for sequences representing all known members of the family *Bornaviridae*[5], resulting in 17,858 non-redundant specific RNA baits and providing a three-fold genome coverage with a length of 80 nt per probe (myBaits® kit with 1–20 K unique baits; Arbor Bioscience, Ann Arbor, MI, USA). The procedure was performed according to the manufacturer's instructions with minor modifications. Briefly, 7 µl of each DNA library were combined with the blocking reagent mix of the kit. After denaturation, 20 µl of a pre-warmed hybridisation mix, including the baits, was added. One volume of mineral oil was used to seal the reaction mix before incubation for 24 h at 65 °C and shaking at 550 rpm in a thermomixer. The aqueous phase was then transferred to a low-binding tube and purified using the binding beads from the myBaits® kit. The enriched target library DNA was finally eluted in 35 µl of 10 mM Tris-HCl, 0.05% Tween-20 solution (pH 8.0-8.5) and amplified in duplicates (16 µl DNA each) using the GeneRead DNA Library L amplification Kit (Qiagen) with 10 cycles (denaturation: 2 min at 98 °C; amplification for 10 cycles: 20 s at 98 °C, 30 s 60 °C, and 30 s at 72 °C; final elongation: 1 min at 72 °C). Subsequently, both duplicates were pooled and purified twice by adding 0.65 or 1.2 volumes of Agencourt AMPure XP Beads (Beckman Coulter) for 500 bp or 200 bp libraries, respectively. Enriched libraries were eluted in 30 µl buffer EB (Qiagen).

### Bioinformatic analysis and quality control of HTS datasets

HTS datasets originating from sequencing with or without BoDV-1 target enrichment were adaptor- and quality-trimmed using the default settings of the 454 software suite (v3.0; Roche) before BoDV-1 reads were identified and extracted by mapping to the BoDV-1 reference sequence NC_001607. Duplicate BoDV-1 reads caused by library amplification were then removed using the SeqKit tool version 0.15.0[63]. Mapping to the reference sequence was repeated for the remaining BoDV-1 reads. In parallel, the remaining BoDV-1 reads were de novo assembled using the 454 software suite and SPAdes v3.13.1[64]. The accuracy of the resulting contigs was checked by comparing the consensus sequences generated by both approaches with each other and by sequence annotation as described below. Discrepancies at single and polynucleotide level were checked by reviewing the raw data quality and data coverage from mapping and assembly. In addition, the effects of discrepancies on gene annotation and frameshifting were checked. If sequence quality and/or coverage was insufficient, Sanger sequencing of RT-PCR amplicons covering the respective positions was performed for confirmation as described below.

### Sanger sequencing

Partial BoDV-1 genome sequences were generated by Sanger sequencing for 43 BoDV-1-positive fresh-frozen brain samples (Table 1), following previously described procedures[25]. Briefly, a 2,272- nt long sequence representing BoDV-1 genome positions 20 to 2,291 (spanning the N, X, P and partial M genes) was determined by sequencing two overlapping PCR products using BoDV-1-specific primers (Supplementary Table 2). The final consensus sequence was generated by assembly of the overlapping raw sequences after trimming of primer-derived sequence ends and manual quality control.

Sanger sequencing was also used to fill gaps or confirm not sufficiently reliable positions in sequences generated by HTS. For this purpose, BoDV-1-specific primer pairs were selected to generate amplicons of approximately 120 to 180 bp length to cover the respective sequence regions. Primer sequences are available upon request.

### Sequence annotation and database submission

BoDV-1 sequences of sufficient length were generated from 136 of 157 selected individuals, including 102 domestic mammals, 25 humans, all seven bicoloured white-toothed shrews as well as the laboratory isolates DessauVac and H24 (Table 1). These sequences included 54 complete coding genomes and 82 sequences covering at least the N-X/P genes (Table 1).

Open reading frames (ORFs) were identified by ORF Finder (implemented in Geneious Prime® 2021.0.1) and verified by sequence alignment to the reference sequence. All sequences generated in this study are available in the INSDC databases under accession numbers OR203629, OR203630, OR468838 to OR468971.

Two previously published isolates (H640 and H3053) were re-analysed, because it was suspected that they may have been interchanged in the original study[38]. As the re-analysis of the original isolates confirmed this suspicion, the corresponding GenBank entries have now been corrected (accession numbers AY374523.2 and AY374537.2).

### Phylogenetic analysis

Phylogenetic analysis of sequences generated in this study was performed together with those publicly available BoDV-1 sequences, which covered at least the N-X/P genes and for which sufficient metadata are available. The used public sequences originated from 55 domestic mammals, 16 human cases, 36 shrews and three laboratory strains isolated from domestic mammals (Table 1). Duplicate sequences originating from the same individual as well as sequences previously identified as laboratory contaminants were excluded from the analysis[7,39,40].

ML trees were constructed individually for 90 complete coding sequences of BoDV-1 genomes and 246 sequences spanning the N-X/P genes (1824 nt, corresponding to genome positions 54 to 1,877). For these analyses, the BoDV-2 No/98 sequence (AJ311524) was used as an outgroup. After sequence alignment using MUSCLE (version 3.8.425)[65], the IQ-TREE software (version 2.2.2.6)[66] was used for phylogenetic reconstruction with automatic model selection. In detail, for complete genome and for N-X/P gene alignments the symmetric model with discrete Gamma model "SYM + G4" and general time reversible model with empirically estimated base frequencies, invariable sites plus discrete Gamma model "GTR + F + I + G4", were selected respectively[67]. Branch support was assessed using SH-aLRT and ultrafast bootstrap tests, with 100,000 replicates for each test[68,69]. For detailed visualisation of the phylogenetic tree without the outgroup, and to display the single phylogenetic clusters and subclusters in separate panels, nodes were extracted using the ggtree R package[70] in R Studio[71] with R v4.0.2[72] exclusively. Heatmap analysis of the genetic cluster similarities was performed using the pheatmap R-package[73].

### Temporal and spatial correlation analysis

Root-to-tip distances and pairwise patristic distances (as nt substitutions per position) were inferred from the ML tree of 247 N-X/P sequences (incl. the BoDV-2 sequence). Temporal correlations were tested by linear regression analysis of root-to-tip distances against year of sampling[74]. Sequences originating from laboratory strains were excluded from the analysis to avoid an impact of adaptive mutations acquired during passaging in cell culture or experimental animals.

IBD analysis was performed, testing the correlation of pairwise patristic distances and geographic distances for all BoDV-1 sequences with available location (n = 238) as well as within the individual BoDV-1 clusters and subclusters. IBD matrix correlations were tested in *R* using the "mantel" function of the "vegan" package (Spearman's rho statistic and 9999 permutations).

### Phylogeographic analysis and determination of BoDV-1 endemic areas

Geospatial data analysis and modelling was performed in R Studio with the packages rnaturalearth[75] and ggplot2[76]. Investigated cases were

mapped according to the geographic location (postal code) available for the metadata assembly. Human cases were visualised no more precise than to the centre of their administrative district (Figs. 1 and 3). Cases with available BoDV-1 sequences were linked to the respective phylogenetic trees by use of colour codes representing phylogenetic clusters and subclusters or subclades (Figs. 1B, C and 3).

Non-parametric KDE was used to visualise spatial distribution patterns of mapped BoDV-1 cases. KDE was performed independently for each phylogenetic cluster or subcluster as well as for sequences of all clusters and subclusters combined. BoDV-1 sequences identified as phylogeographic outliers by using the outlier definition described in detail in the results section (presence of no other BoDV-1 N-X/P sequence with ≥98.6% nt identity within a distance of ≤37.9 km) were excluded from the KDE.

The two-dimensional KDE, implemented in ggplot as the "stat_density_2d" function[77], was used with a polygon as the bounding box of estimated endemic regions. To smoothen the polygon, $n = 100$ grid points were defined in each direction. A low bandwidth (h) was set empirically for both approaches in order to minimise the extent of the estimated areas beyond the confirmed cases (subcluster 1A: 1; subcluster 1B: 0.5; cluster 2: 0.6; cluster 3: 0.75; cluster 4: 0.65; combination of all clusters: 1.0).

### Ethics statement

Ethical approval of the analysis of archived human samples was obtained from the local ethical commission of the Faculty for Medicine, University of Regensburg (ref. no. 18-1248-101), the Technical University Munich (577/19 S), the Ludwig-Maximilians University Munich (23-0267) and the Medical Board of Hamburg (PV5616). Samples of BoDV-1-positive bicoloured white-toothed shrews were obtained from an ongoing large-scale small mammal screening study (Haring et al., manuscript in preparation). Shrew KS20/0026 originated from a project that was commissioned by the Federal Environment Agency as part of the Environmental Research Plan (Research Code 3718 48 4250; animal ethics permit: 42502-2-1548 Uni Leipzig) and was financed with federal funds. All other shrew carcasses included in this study were found dead or preyed by cats. Samples from domestic mammals originated from diagnostic necropsies. No living animals were handled or killed for the purpose of this study.

### Reporting summary

Further information on research design is available in the Nature Portfolio Reporting Summary linked to this article.

## Data availability

All novel BoDV-1 sequences are available from the INSDC databases under accession numbers OR203629, OR203630, OR468838 to OR468971 (https://www.ncbi.nlm.nih.gov/nucleotide/). Accession numbers and metadata are listed in a separate file (Supplemental Data Set 1). BoDV-1 sequences of reanalysed previously published isolates (H640 and H3053) are available under accession numbers AY374523.2 and AY374537.2. Source data are provided with this paper.

## Code availability

Genome reconstruction (mapping, assembly) was performed using 454 software suite and SPAdes v3.13.1 51[64]. Sequence analysis was performed using Geneious Prime 2021.0.1; Biomatters, Auckland, New Zealand. Phylogenetic analysis was performed for complete-coding and partial sequences (N-X-P genes) using IQ-TREE software (version 2.2.2.6)[66], based on a MUSCLE-Alignments[65]. Heatmap analysis was performed using pheatmap R-package[73]. Root-to-tip distances were calculated from the ML tree of N-X/P sequences using TempEst[74]. Isolation by distance analysis was perfomed using the "mantel" function of the "vegan" R-package. Geospatial data analysis and modelling was performed in R Studio (R v4.0.2) with the packages rnaturalearth[75]

and ggplot2[76]. Non-parametric two-dimensional kernel density estimation was used to visualise spatial distribution patterns of mapped BoDV-1 cases, using "stat_density_2d" function in ggplot.

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

## Acknowledgements

We would like to thank Patrick Zitzow, Kathrin Steffen, Weda Hoffmann, Lukas Wessler, Jessica Geers and Elsbeth Keller-Gautschi for their outstanding technical assistance. Brigitte Böhm, Eva Kappe (both Poing, Germany), Wolfram Breuer, Melanie Bühler, Anne Kupca (all Oberschleissheim, Germany), Gesine Buhmann, Karin Weber (both Munich, Germany), Klaus-Jürgen Danner (Freiburg, Germany), Vanessa Franzen (Munich, Germany), Sascha Gerst (Rostock, Germany), Ernst Großmann (Aulendorf, Germany), Wolfram Haider (Berlin, Germany), Anja Heinrich, Claudia Kiesow (both Stendal, Germany), Christian Imholt, Jens Jacob, Philipp Koch (Münster, Germany), Andrea Konrath, Martin Pfeffer (Leipzig, Germany), Martin Peters (Arnsberg, Germany), Dietrich Pöhle (Dresden, Germany), Ingo Schwabe (Fellbach, Germany), Christoph Schulze (Frankfurt/Oder, Germany), Herbert Weissenböck (Vienna, Austria) and Eva-Maria Wittauer (Bad Kissingen, Germany) submitted diagnostic material from confirmed or suspected cases of Borna disease or BoDV-1-infected shrews. Furthermore, we like to thank all veterinarians and physicians treating the analysed animals and human patients, respectively. We are grateful to Sybille Herzog (Giessen, Germany) for providing BoDV-1 isolates for re-sequencing, Sven Springer (IDT Biologika, now Ceva Santé Animale, Dessau-Rosslau, Germany) for kindly providing a vial of the bornavirus live vaccine 'Dessau' and Christiane Herden (Giessen, Germany) for providing the laboratory strain H24. We like to thank Dirk Höper for providing funding, technical supervision and advice as well as for critically discussing the data analysis and the manuscript. This work was supported by the Federal Ministry of Education and Research within the research consortium "ZooBoCo" (Grant no. 01KI1722 and 01KI2005 donated to Martin Beer, Dirk Höper, Timo Homeier-Bachmann, Kirsten Pörtner, Dennis Rubbenstroth, Dennis Tappe and Rainer G. Ulrich) and the projects "ZooKoInfekt" (01KI1903B; Rainer G. Ulrich and Dennis Rubbenstroth) and "Bornavirus - Focal Point Bavaria" (01KI2002; Barbara Schmidt). Friederike Liesche-Starnecker received funding from the German Research Foundation (DFG; no. 504757758).

## Author contributions

D.R. initiated and conceptualised the study. R.D., J.K., V.R., F.L.S., R.U., J.F., K.M., F.H., T.S., D.N., M.M., A.N.J., M.S., M.Ba., H.H.N., B.S., V.C.H., K.P., C.F., L.M., J.H., W.B., N.N., J.S. and R.G.U. provided samples together with clinical, histopathological and initial laboratory diagnosis and assembly of metadata. A.Eb., P.D.S., D.R., D.T., D.C., K.K., A.En., J.K., K.S., B.H. and D.N. performed PCR assays and sequencing. D.R., A.Eb., F.P. and T.H.B. analysed and visualised the data. R.G.U. and M.Be. supervised parts of the study. D.R. and A.Eb. wrote and D.R. finalised the manuscript. The manuscript was critically reviewed through the contributions of all authors.

## Funding

## Competing interests

The authors declare no competing interests.

## Additional information

Arnt Ebinger[1], Pauline D. Santos [1], Florian Pfaff [1], Ralf Dürrwald [2], Jolanta Kolodziejek [3], Kore Schlottau [1], Viktoria Ruf[4], Friederike Liesche-Starnecker [5,6], Armin Ensser [7], Klaus Korn [7], Reiner Ulrich[8], Jenny Fürstenau [9], Kaspar Matiasek [10], Florian Hansmann [8,11], Torsten Seuberlich[12], Daniel Nobach [13,14], Matthias Müller[15], Antonie Neubauer-Juric [16], Marcel Suchowski[8,16], Markus Bauswein[17], Hans-Helmut Niller[18], Barbara Schmidt[17], Dennis Tappe[19], Daniel Cadar[19], Timo Homeier-Bachmann [20], Viola C. Haring [21], Kirsten Pörtner [22], Christina Frank [22], Lars Mundhenk [9], Bernd Hoffmann [1], Jochen Herms [4], Wolfgang Baumgärtner [11], Norbert Nowotny [3,23], Jürgen Schlegel[24], Rainer G. Ulrich [21], Martin Beer [1] & Dennis Rubbenstroth [1] ✉

[1]Institute of Diagnostic Virology, Friedrich-Loeffler-Institut, Greifswald-Insel Riems, Germany. [2]Robert Koch Institute, Department of Infectious Diseases, Unit 17 Influenza and Other Respiratory Viruses, National Reference Centre for Influenza, Berlin, Germany. [3]Institute of Virology, University of Veterinary Medicine Vienna, Vienna, Austria. [4]Center for Neuropathology and Prion Research, Faculty of Medicine, Ludwig-Maximilians-Universität München, Munich, Germany. [5]Department of Neuropathology, Pathology, Medical Faculty, University of Augsburg, Augsburg, Germany. [6]Pathology, Medical Faculty, University of Augsburg, Augsburg, Germany. [7]Institute of Virology, University Hospital Erlangen, Friedrich-Alexander Universität Erlangen-Nürnberg (FAU), Erlangen, Germany. [8]Institute of Veterinary Pathology, Faculty of Veterinary Medicine, Leipzig University, Leipzig, Germany. [9]Institute of Veterinary Pathology, Freie Universität Berlin, Berlin, Germany. [10]Section of Clinical & Comparative Neuropathology, Centre for Clinical Veterinary Medicine, Ludwig-Maximilians-Universität München, Munich, Germany. [11]Department of Pathology, University of Veterinary Medicine Hannover, Hannover, Germany. [12]Division of Neurological Sciences, Vetsuisse Faculty, University of Bern, Bern, Switzerland. [13]Institute of Veterinary Pathology, Justus-Liebig-University Giessen, Giessen, Germany. [14]Chemical and Veterinary Analysis Agency Stuttgart (CVUAS), Fellbach, Germany. [15]Bavarian Health and Food Safety Authority, Erlangen, Germany. [16]Bavarian Health and Food Safety Authority, Oberschleißheim, Germany. [17]Institute of Clinical Microbiology and Hygiene, Regensburg University Hospital, Regensburg, Germany. [18]Institute for Medical Microbiology, Regensburg University, Regensburg, Germany. [19]Bernhard Nocht-Institute for Tropical Medicine, Hamburg, Germany. [20]Institute of Epidemiology, Friedrich-Loeffler-Institut, Greifswald-Insel Riems, Germany. [21]Institute of Novel and Emerging Infectious Diseases, Friedrich-Loeffler-Institut, Greifswald-Insel Riems, Germany. [22]Robert Koch Institute, Department of Infectious Disease Epidemiology, Berlin, Germany. [23]Department of Basic Medical Sciences, College of Medicine, Mohammed Bin Rashid University of Medicine and Health Sciences, Dubai, United Arab Emirates. [24]Department of Neuropathology, School of Medicine, Institute of Pathology, Technical University Munich, Munich, Germany. ✉e-mail: Dennis.Rubbenstroth@fli.de

