## [Peer Review File · Nature Communications]

Lethal Borna disease virus 1 infections of humans and animals
– in-depth molecular epidemiology and phylogeographyREVIEWER COMMENTS

Reviewer #1 (Remarks to the Author):

The research article entitled "Lethal Borna disease virus (BoDV-1) infections of humans and animals- in depth molecular epidemiology and phylogeography" by Ebinger et al., constitutes a multidisciplinary, multi institutional collaborative research effort that looks to unveil details on the spillover infection patterns of BoDV-1 from its putative reservoir host the bicolored white toothed shrew (*Crocidura leucodon*) to sympatric domestic animals and humans across putative endemic locations in Germany, Switzerland, Liechtenstein and Austria. In addition, authors also investigated the geographic distribution, as well as the spatial temporal (1964-2023) dissemination patterns to better understand transmission and dissemination trends. The sampling, metadata collection approaches, laboratory and analytical methods used were robust and sounded through the entire investigation, allowing excellent data integration for robust phylogeographic inferences. Result's description is thorough and comprehensive with exquisite details provided to understand operative concepts that helped established objective definitions of endemicity. Discussion is compiling and complete, authors explored all potential scenarios that explained their data and results, as well as highlighted with caution alternative hypothesis to explain outlier data. In general, this research provides the most comprehensive and robust phylogeographic insight on BoDV-1 across putative endemic regions in Europe, as well as unveils important aspects on potentially seasonal spillover infection trends to humans, domestic and some captive wild animals from its most likely reservoir host the white-toothed shrew. Authors also made some conservative efforts to suggest some hypotheses that may explain all potential gaps or weaknesses in their research with which the highlight big avenues for future research.

Minor comments/suggestions

Lines 319-320 Authors did an outstanding work assembling such a complete descriptive epidemiological profile of human deaths/cases associated with BoDV-1 infection. However, I would like to ask authors if there is some experimental data (animal models) on infectious dose and route in borna disease and how that may impact the better understanding of incubation periods in borna disease.

I also wonder whether case fatalities could also be related to decadent (elderly), unmaturred or immunocompromised immune systems in very young or elderly individuals given the wide age group BoDv-1 may affect. Were there any age trends in affected individuals suggesting any of the above hypotheses?

Line 572 Passive vector or fomites. Contaminated feed. Authors use the word feed multiple times in the context of food. I think this may be a typo that requires to be corrected throughout the entire manuscript.

General comment.

I think it would be beneficial having some authors insight on the importance of population genetics data on the putative BoDV-1's reservoir host (white-toothed shrew) to re-enforce phylogeographic inferences. Given that most of the discussion focuses on data generated from the virus. I think that coupled population genetics studies of infected and uninfected, white-toothed shrews across putative endemic and nonendemic areas would help to understand endemicity and temporal dissemination dynamics.

Reviewer #2 (Remarks to the Author):

The authors report on the zoonotic transmission of BoDV-1, a virus associated with Borna disease, a progressively and fatal neurologic disease of humans and animals.

Major comments

- The methods section should be revised to indicate sample selection and metadata selection. For examples details of sample selection and acquisition are mixed in the sub-headings, line 179-184 could be described elsewhere and only focus on how the high-throughput sequencing was conducted and carried out.

- The section from line 193 to 201 comprises of bioinformatics analysis and should be in its own sub-heading.
- The authors report that BoDV-1 reads were assembled with SPAdes version 3.13.1, this version was released in 2019 (>4 years ago), could they justify the use of this version of the assembly given that SPAdes has had several releases from 2019.
- The authors claim to have carried out manual sequence curation, but they do not provide details of what was and how the manual curation was conducted and raises questions of reproducibility of the sequence data. Could you provide exactly what was modified and how and for what reason.
- A summary diagram of the sequencing approach would be useful to clarify the steps and which samples were sequenced or enriched and for what reasons.
- The section on definition and further analysis of phylogeographic outlines could be more appropriately titles and the text could be more succinct in a way that communicates the most important findings.
- The authors mention phylogeographic analysis in the results section, however it is not clear in the methods section how this was conducted or carried out. It would be useful if the author could revise the appropriate section in the methods and provide clear headings
- Results section from line 495-498 could go the methods section. This could be revised to separate the methods from the results, as written it is difficult to follow.

Minor comments

- line 57-58 - provide the number of cases that were diagnosed. Was this clinical diagnosis?
- line 63 - the authors report 136 new complete and partial genomes. Kindly provide the number of complete and also the partial genomes. Clarify what partial in this case means in terms of genome completeness.
- line 108, provide the number of the archived samples and those of the current samples.

Line332-335 , This paragraph is not very clear. Consider revising. When you say "...136 individuals including 102 domestic animals , 25 humans , all seven shrews .." This is statement is ambiguous as to the total number of samples and which is from which.?

Line 338 starts with "including additional" It is not clear this is additional to what? Consider revising the sentence to add more clarity to the readers.

The statements in lines 332-343 could be supported by a diagram that illustrates the flow and which samples are from which sources.

Line 467: is this sentence part of the previous sentence? If so, this needs to be grammatically joined up with the previous sentence. Otherwise it sounds like an incomplete sentence. And the same applies to the sentence starting at line 469.

Line 477; remove the work "clearly."

Line 478: I am not sure if something can be almost identical. It is either identical or not. Could the authors clarify what they mean here.

Line 496 ; KDE abbreviation should be clarified in full.

Line 524 remove the word "stably"

Line 540: provide a reference after "previously" to support the argument provide here.

Reviewer #3 (Remarks to the Author):

I generally feel positive about the manuscript. There has been a great deal of work compiling the data and this is great to see alongside generally sound phylogenetic work. My main concern is around the description of the process used to identify phytogeographic outliers. I think this is primarily a presentation issue rather than a critical flaw in the analysis though I think it should be addressed with some care before publication.

Firstly, please properly explain the process used to identify phylogeographic outliers in the Methods before describing the KDE rather than just referring forwards to the Results when describing the KDE. If a second set of criteria were used to identify potential outliers that appear in brackets (Y & Z), state these in the Methods too. I can see these two cases referred to in the footnote to extended data table 3 but it should be properly explained in the Methods.

I don't really understand why the authors describe calculating patristic distances and plot these on the x axis of extended data figure 8 and Figure 4, yet use nucleotide similarity in the criteria for identification of phylogeographic outliers. Why not use patristic distance in the outlier definition? Could this be done retrospectively? If patristic distance was used in the definition, the cut-off used could be visualised on these x-axes. If patristic distances and the phylogeny are not used to identify outliers, the term phylogeographic outlier may be misleading.

Is the one exception mentioned on line 417 the sequence denoted elsewhere as F? If so, why not label it as F on line 417. More generally, it is not useful to lump in F with the other outliers which are inferred to arise through longer distance human movements. F is a different kind of outlier – it is not sampled in an unexpected area but does have a higher genetic distance to other sequences. I can't see the advantage in labelling these in the same way. Also, why should F be excluded from the KDE?

There seems to be an inconsistency in the following which relates to the definition of phylogeographic outlier:

Line 419: 'with the 90th percentile at 37.9 km'

Line 1050: 'Broken horizontal lines represent the 90th percentile (39.8 km)'

Figure 4 has issues, though I really like Extended data figure 8. I think having the left and right panels for each cluster with and without phylogeographic outliers in Extended data figure 8 is visually striking. If I may offer a suggestion, I would incorporate a version of Extended data figure 8 'all sequences' (but without the grey inter-cluster comparisons) into Fig 4. This might help the reader understand the process of what the authors have done. In Extended data figure 8 itself, it would be useful to include N in each of the right and left plots so that the number of outliers excluded in each case can be inferred.

Given the KDE is not mentioned until the last paragraph of the results section and is reliant on the identification of phylogeographic outliers, it would make more sense to have the KDE visualisation currently in Fig 1 parts C and D as part of Fig 4.

Further issues/queries associated with Fig 4

The significance of Y having brackets (while N and F do not) is not stated in the legend.

It therefore does not make sense why is Y described in the legend while N and F are not.

'Sequence Z21_0139 (Y) is marked as a potential outlier due to its close genetic relation to animal sequences in more than 200 km distance' – This is confusing as having noted the criteria for outlier definition, I would have taken from the paper that more accurately, it is the absence of geographically close genetic relations rather than the presence of geographically distant close genetic relations that the described method for outlier detection relies on. However, I think the point is that Y would not be classified as an outlier using the definition elsewhere in the paper - It is not clear from the legend that the standard definition is being overridden.

Were there any instances where a human BoDV-1 sequence had >1 sequence with equally minimal patristic distance? If so, to which of these would the spatial distance be plotted in Fig 4?

There is a green point at around 0.010 on the x-axis that appears above the dashed line at 39.8km. Why is this not identified as an outlier using the stated 37.9km cutoff?

Similarly, it is confusing to see F identified as a phylogeographic outlier when it falls on the median line. The reader currently must jump around the manuscript somewhat to figure out why this is the case, which is not ideal.

Minor

Line 60: clarify wild shrews?

Line 85-86. 'Within days to months' relative to what.. exposure, onset of fever, hospitalisation?

Line 88. Reference 15 duplicated

Line 110. Shrew sample origins could be briefly outlined.

Line 258-259. Substitution model components should be written out before acronyms at first use. 'General time reversible model of nucleotide substitution'. 'a proportion of invariant sites' etc. I'm not familiar with 'F' in the N-X/P model, does that denote estimated base frequencies?

Line 260-261. Not clear what was done with ggtree here. Is that simply the exclusion of the outgroup sequence for visualisation? Ideally, the tree including the outgroup should be included in the supplementary figures.

Line 266-267. Please state reason for exclusion of laboratory strains in the text. Was this because dates were unknown? Or to guard against lab-adaptative mutations that may have been acquired since original sampling date?

Line 340. 'Clusters 1 to 4 and subclusters 1A and 1B' would be more accurate. The presence of subclusters within it does not negate the presence of cluster 1.

General authors' comments:

We thank the reviewers for their detailed and constructive comments on our work, which helped to further improve our manuscript. We have provided a point-by-point response to each comment below.

Please note, that in contrast to the original manuscript, the Methods section has now been moved to the end of the manuscript in accordance with the format of Nature Communications. The line numbers in our responses are referring to the revised manuscript and may thus differ from the line numbers used by the reviewers.

REVIEWER COMMENTS

Reviewer #1 (Remarks to the Author):

The research article entitled “Lethal Borna disease virus (BoDV-1) infections of humans and animals- in depth molecular epidemiology and phylogeography” by Ebinger et al., constitutes a multidisciplinary, multi institutional collaborative research effort that looks to unveil details on the spillover infection patterns of BoDV-1 from its putative reservoir host the bicolored white toothed shrew (*Crocidura leucodon*) to sympatric domestic animals and humans across putative endemic locations in Germany, Switzerland, Liechtenstein and Austria. In addition, authors also investigated the geographic distribution, as well as the spatial temporal (1964-2023) dissemination patterns to better understand transmission and dissemination trends. The sampling, metadata collection approaches, laboratory and analytical methods used were robust and sounded through the entire investigation, allowing excellent data integration for robust phylogeographic inferences. Result’s description is thorough and comprehensive with exquisite details provided to understand operative concepts that helped established objective definitions of endemicity. Discussion is compiling and complete, authors explored all potential scenarios that explained their data and results, as well as highlighted with caution alternative hypothesis to explain outlier data. In general, this research provides the most comprehensive and robust phylogeographic insight on BoDV-1 across putative endemic regions in Europe, as well as unveils important aspects on potentially seasonal spillover infection trends to humans, domestic and some captive wild animals from its most likely reservoir host the white-toothed shrew. Authors also made some conservative efforts to suggest some hypotheses that may explain all potential gaps or weaknesses in their research with which the highlight big avenues for future research.

Minor comments/suggestions

Lines 319-320 Authors did an outstanding work assembling such a complete descriptive epidemiological profile of human deaths/cases associated with BoDV-1 infection. However, I would like to ask authors if there is some experimental data (animal models) on infectious dose and route in borna disease and how that may impact the better understanding of incubation periods in borna disease.

Response: Experimental models in a broad range of mammalian species indeed demonstrated that route of infection as well as dose may affect the incubation period. We have added this information (line 84) and cited a very detailed and extensive study performed in the well-established rat model (Carbone et al., JVI, 1987).

I also wonder whether case fatalities could also be related to decadent (elderly), unmatured or immunocompromised immune systems in very young or elderly individuals given the wide age group BoDV-1 may affect. Were there any age trends in affected individuals suggesting any of the above hypotheses?

Response: So far, no obvious age trends have been observed, as patients of all age groups have been affected. A previous study of Pörtner et al., EMI, 2023 analysing 20 cases in detail found the age groups of 10-29 and 70-79 years of age slightly overrepresented as compared to the general population of the respective administrative districts. However, this may have also reasons other than maturity of the immune system or other medical conditions. Pörtner et al. found living at the edge of small villages as the most significant risk factor for acquiring BoDV-1 encephalitis (see line 459-461). It may be conceivable to assume that older persons as well as families with children may be overrepresented in such locations, as compared to the middle-aged working population without children, that may be more likely to live in towns/cities or flats located more in the centre of their settlements. Furthermore, the age groups <10 years and >80 years were completely absent from their dataset, rather not suggesting maturity of the immune system being an important factor. Moreover, BoDV-1-induced encephalitis is known to be the result of T lymphocyte-mediated immunopathogenesis (see lines 80) and in animal models newborn and/or immunocompromised individuals were rather protected against encephalitis (despite establishing a persistent infection of the CNS).

In principal, it would be possible to perform statistical analysis of the age distribution also for our dataset. However, since we did not perform a detailed follow-up of each case, as performed by Pörtner et al., interpretation of the results may be even more difficult. Furthermore, these aspects were not exactly in the focus of our study and the additional analyses and their thorough discussion would further increase the length and complexity of this manuscript. We therefore suggest to leave the suggested analyses for further follow-up studies focussing on the human cases.

Line 572 Passive vector or fomites. Contaminated feed. Authors use the word feed multiple times in the context of food. I think this may be a typo that requires to be corrected throughout the entire manuscript.

Response: The term “feed” is as established term used in animal nutrition, while “food” is used mainly for humans. Since we are referring to animals in both instances in which we used the term “feed” (lines 423 and 485), we have not changed it.

General comment.

I think it would be beneficial having some authors insight on the importance of population genetics data on the putative BoDV-1’s reservoir host (white-toothed shrew) to re-enforce phylogeographic inferences. Given that most of the discussion focuses on data generated from the virus. I think that coupled population genetics studies of infected and uninfected, white-toothed shrews across putative endemic and nonendemic areas would help to understand endemicity and temporal dissemination dynamics.

Response: We agree that these are important aspects to better understand the epidemiology and the surprisingly restricted endemic area of BoDV-1. While this study focusses mainly on BoDV-1 infection on non-reservoir hosts (domestic mammals and humans), a study analysing the distribution of BoDV-1 in shrews has been conducted in parallel, with the manuscript currently being prepared (Haring et al., manuscript in preparation). As part of this study, also investigations into the population genetics of the reservoir will be performed.

Reviewer #2 (Remarks to the Author):

The authors report on the zoonotic transmission of BoDV-1, a virus associated with Borna disease, a progressively and fatal neurologic disease of humans and animals.

Major comments

- The methods section should be revised to indicate sample selection and metadata selection. For examples details of sample selection and acquisition are mixed in the sub-headings, line 179-184 could be described elsewhere and only focus on how the high-throughput sequencing was conducted and carried out.
- The section from line 193 to 201 comprises of bioinformatics analysis and should be in its own sub-heading.

Response: The selection of samples for sequencing has been summarized under a new subheading prior to the different sections describing the sequencing

procedures (lines 569-579), Likewise, analysis and quality control of high-throughput sequencing (HTS) data has been moved to a separate subsection (lines 608-621).

- The authors report that BoDV-1 reads were assembled with SPAdes version 3.13.1, this version was released in 2019 (>4 years ago), could they justify the use of this version of the assembly given that SPAdes has had several releases from 2019.

Response: When the study was initiated in 2019, SPAdes version 3.13.1 was up-to-date. Since version 3.13.1, nine additional versions have been released until now, with the current version being 3.15.5 (released 14 July 2022). A review of the changelog ([spades/ assembler/ changelog.html](https://spades.github.io/assembly/changelog.html) at [spades_3.15.5 · ablab/spades · GitHub](https://github.com/ablab/spades)) reveals the introduction of new functions, bug fixes, and optimizations. However, none of these changes appear to significantly impact the fundamental assembly algorithm or raise concerns regarding the reproducibility of results obtained using older versions. Therefore, we have determined that there is no immediate necessity to update the software version during the study. We chose to maintain continuity by keeping version 3.13.1 until the completion of the study.

- The authors claim to have carried out manual sequence curation, but they do not provide details of what was and how the manual curation was conducted and raises questions of reproducibility of the sequence data. Could you provide exactly what was modified and how and for what reason.

Response: During the quality analysis phase, the comparison of the three different assembly approaches revealed isolated discrepancies in genomes at both the single and polynucleotide level. Detailed analysis was then performed on these specific sites, including a detailed review of the raw data and data coverage using mapping of reads back to the assembled genome. In addition, the impact of these discrepancies on gene annotation and frameshifting was assessed to identify possible errors among the discrepancies.

In cases where there was any doubt that the data coverage for some positions in the genome was insufficient, Sanger sequencing or the myBaits BoDV-1 sequencing method were used to verify the integrity of these sites. This approach was intended to ensure the accuracy and reliability of the genomic data generated by the assembly processes. We clarified this in lines 617 to 621.

- A summary diagram of the sequencing approach would be useful to clarify the steps and which samples were sequenced or enriched and for what reasons.

Response: A diagram summarizing the analysis workflow is now included as Extended data figure 10. Furthermore, information on the applied sequencing and analysis steps for each sequenced sample is now provided in Extended Data Table 5.

- The section on definition and further analysis of phylogeographic outlines could be more appropriately titled and the text could be more succinct in a way that communicates the most important findings.

Response: The section has now been divided into two separate subsections with the titles 'Definition of phylogeographic outliers' (lines 253-272) and 'Detailed characteristics of defined phylogeographic outliers' (lines 273-304). While the first section describes the definition criteria, the second subsection summarizes the metadata available on these outliers to give the reader an impression of the various potential reasons for cases being outliers as well as of the limitations of the available metadata. This information is the basis for the subsequent discussion (lines 408-432).

- The authors mention phylogeographic analysis in the results section, however it is not clear in the methods section how this was conducted or carried out. It would be useful if the author could revise the appropriate section in the methods and provide clear headings

Response: Further information of the phylogeographic analysis has been added and the title of the respective results section has been changed to 'Phylogeographic analysis and determination of BoDV-1 endemic areas' (lines 676-692).

- Results section from line 495-498 could go to the methods section. This could be revised to separate the methods from the results, as written it is difficult to follow.

Response: The information, that phylogeographic outliers were excluded from the KDE analysis is already available in the respective subsection of the Methods section (lines 683-687). Since the Methods section has now been moved to the end of the manuscript, in accordance with the Nature Communications format, we think it is important for the reader to get this crucial information also at the beginning of the respective Results subsection.

Minor comments

- line 57-58 - provide the number of cases that were diagnosed. Was this clinical diagnosis?

Response: The number of cases are already provided in the following sentence for each individual category. The initial diagnosis was based on clinical and/or histopathological characteristics. This information has now been added to lines 59-60.

-line 63 - the authors report 136 new complete and partial genomes. Kindly provide the number of complete and also the partial genomes. Clarify what partial in this case means in terms of genome completeness.

Response: Since the abstract of the article should be short and concise, we suggest to not add these details to the abstract. The reader will find the respective information in the main text as well as in Table 1.

-line 108, provide the number of the archived samples and those of the current samples.

Response: This final paragraph of the Introduction is intended to provide a general overview of the study without going into details. Detailed information (e.g. the number of fresh-frozen and archived FFPE samples) is provided in the Methods section and in Table 1.

Line 332-335 , This paragraph is not very clear. Consider revising. When you say “...136 individuals including 102 domestic animals , 25 humans , all seven shrews ..” This is statement is ambiguous as to the total number of samples and which is from which.?

Response: An introductory sentence has now been added to this section to better explain what has been done (lines 169-171).

Line 338 starts with “including additional” It is not clear this is additional to what? Consider revising the sentence to add more clarity to the readers.

Response: The sentence has been rephrased to clarify that the additional sequences were derived from public databases (lines 177-178).

The statements in lines 332-343 could be supported by a diagram that illustrates the flow and which samples are from which sources.

Response: The requested information is already provided in Table 1. An additional reference to Table 1 has been added for clarification (line 178).

Line 467: is this sentence part of the previous sentence? If so, this needs to be grammatically joined up with the previous sentence. Otherwise it sounds like an incomplete sentence. And the same applies to the sentence starting at line 469.

Response: Each of these sentences is a grammatically complete sentence. To avoid long interlaced sentences, we suggest to keep them separate.

Line 477; remove the work “clearly.”

Response: The word “clearly” has been removed (line 325).

Line 478: I am not sure if something can be almost identical. It is either identical or not. Could the authors clarify what they mean here.

Response: The sentence was changed to “Its sequence possessed 99.9% nt sequence identity to the sequence of the vaccine strain ‘DessauVac’ (Extended Data Figure 5C)” (line 326).

Line 496 ; KDE abbreviation should be clarified in full.

Response: The definition of KDE has now been added to this section, as it is now the first occurrence of the abbreviation (lines 345-346).

Line 524 remove the word “stably”

Response: The word “stably” was removed (line 372).

Line 540: provide a reference after “previously” to support the argument provide here.

Response: The sentence was rephrased to list the cited references behind the word “previously” (lines 379-380).

Reviewer #3 (Remarks to the Author):

I generally feel positive about the manuscript. There has been a great deal of work compiling the data and this is great to see alongside generally sound phylogenetic work. My main concern is around the description of the process used to identify phylogeographic outliers. I think this is primarily a presentation issue rather than a critical flaw in the analysis though I think it should be addressed with some care before publication.

Firstly, please properly explain the process used to identify phylogeographic outliers in the Methods before describing the KDE rather than just referring forwards to the Results when describing the KDE. If a second set of criteria were used to identify potential outliers that appear in brackets (Y & Z), state these in the Methods too. I can see these two cases referred to in the footnote to extended data table 3 but it should be properly explained in the Methods.

Response: After moving the Methods section to the end of the manuscript to adhere to the format of Nature Communications, the details of the outlier identification described in the results section are now already known to the reader

when reaching the Methods section. We therefore think that referring to the results section for more details rather than duplicating them is now appropriate.

We agree that calling the two cases Y/Z “potential outliers” was misleading. We have now labelled them as cases #1 and #2 and refer to them as ‘additional cases with potentially aberrant infection sources’ throughout the manuscript (lines 331, 1070, 1089; Extended data table 3) to clarify that they were not treated as phylogeographic outliers during the subsequent analyses (see also further comment to Figure 4 below).

I don't really understand why the authors describe calculating patristic distances and plot these on the x axis of extended data figure 8 and Figure 4, yet use nucleotide similarity in the criteria for identification of phylogeographic outliers. Why not use patristic distance in the outlier definition? Could this be done retrospectively? If patristic distance was used in the definition, the cut-off used could be visualised on these x-axes. If patristic distances and the phylogeny are not used to identify outliers, the term phylogeographic outlier may be misleading.

Response: We deliberately intended to keep the outlier definition as simple as possible, so that also other groups and diagnostic laboratories could easily adapt it after adding one or few additional BoDV-1 sequences, without requiring a phylogenetic analysis of the whole dataset. Furthermore, nucleotide sequence identities of two given sequences are fixed values, whereas tree topologies and thereby patristic distances may slightly vary when adding or removing sequences, potentially leading to a situation in which a sequence is just above the cut-off in one analysis and just below in the next.

However, we have now performed a comparative outlier analysis using patristic distance instead of sequence identities. Using the same approach, we defined a patristic distance of 0.015 and a spatial distance of 37.9 km as cut-offs. All 24 outliers of our outlier analysis were confirmed by the comparative analysis, whereas additional outliers were not identified, demonstrating the robustness of our approach. We have added the comparative analysis to the supplemental material (Extended Data Figure 10), mentioned the identical results in lines 267-269) and discussed the rationale for using nucleotide sequence identities in lines 400-403. Due to the striking agreement of both approaches, we think that the use of the term “phylogeographic outlier” is justified.

Plotting patristic distances vs. spatial distances is an established procedure of IBD analysis (Supplemental figure 8), so that we used patristic distances for this analysis. For Figure 4 (now Figure 5), plotting the genetic distance was not essential to show that most human sequences were located not far from their next relatives. However, we found the genetic information useful to illustrate that those with higher spatial distance often did not have a particularly close relative available in our dataset. We agree, that this information could have been provided

by using patristic distances (to be consistent with the IBD analysis) as well as nucleotide sequence identities (to be consistent with the outlier definition).

Is the one exception mentioned on line 417 the sequence denoted elsewhere as F? If so, why not label it as F on line 417. More generally, it is not useful to lump in F with the other outliers which are inferred to arise through longer distance human movements. F is a different kind of outlier - it is not sampled in an unexpected area but does have a higher genetic distance to other sequences. I can't see the advantage in labelling these in the same way. Also, why should F be excluded from the KDE?

Response: *The case has now been labelled as F already in line 260.*

The defined outliers are not necessarily the result of human (or animal) movements. Some of them may as well result from incorrect location, while others may indicate a genuine occurrence of BoDV-1 in their region, but another closely related sequence to confirm this has not been sampled, yet. We agree, that this may well be the case for outlier F, but further confirmation is missing. The same applies also to other outliers, such as two horses from northern Germany, that were reported to have never left their region (or even stable) of origin, raising the question, whether they indeed represent an endemic occurrence that is just missing confirmation by a second sampled case. This has been discussed in lines 408-426. Since case F does meet the criteria for our outlier definition and the potential reasons are similar to those for other outliers, we think that creating a separate category and nomenclature will more likely cause confusion rather than clarification.

The KDE analysis is deliberately based only on those cases, for which indication of an endemic occurrence has been confirmed with at least some degree of reliability by at least a second sampled case of a related sequence in the same region (thus by not meeting our outlier criteria), as discussed in lines 392-405. Since outlier F, as well as e.g. the horses from northern Germany mentioned above, are lacking this confirmation, they needed to be removed from the KDE analysis. We are well aware, that this may lead to our KDE analysis not representing all actual endemic areas and that some of the current outliers may be confirmed to represent endemicity by further cases diagnosed in the future. This has been discussed in lines 424-432.

There seems to be an inconsistency in the following which relates to the definition of phylogeographic outlier:

Line 419: 'with the 90th percentile at 37.9 km'

Line 1050: 'Broken horizontal lines represent the 90th percentile (39.8 km)'

Response: *These two numbers do actually represent the 90th percentile of two different datasets: the 37.9 km is the 90th percentile of the minimal spatial distances of each sequence to any other sequence with $\geq 98.6\%$ sequence identity*

and thus representing the cut-off of our outlier definition. The broken horizontal line in Figure 4 represents the 90th percentile of the spatial distances of only the human-derived sequences to their closest relatives. It is only used as a descriptive tool to demonstrate that the majority of patients has likely been exposed in their home region.

For clarification, we have added the information that the broken line at 39.8km represents the 90th percentile ‘of the presented dataset’ to the legend of Figure 4 (line 1104).

Figure 4 has issues, though I really like Extended data figure 8. I think having the left and right panels for each cluster with and without phylogeographic outliers in Extended data figure 8 is visually striking. If I may offer a suggestion, I would incorporate a version of Extended data figure 8 ‘all sequences’ (but without the grey inter-cluster comparisons) into Fig 4. This might help the reader understand the process of what the authors have done.

Response: We have added the suggested Figure summarizing only the intra-cluster comparisons to the main manuscript as a new Figure 3. We have not incorporated it into the former Figure 4 (now Figure 5), since both are representing very different data sets and approaches. A combination of both might cause confusion rather than help the readers to understand the process. It has to be noted that the new Figure 3 is only an illustrative summary of the IBD analysis shown in Extended Data Figure 8, but has no analytic value itself.

In Extended data figure 8 itself, it would be useful to include N in each of the right and left plots so that the number of outliers excluded in each case can be inferred.

Response: We have added N to each panel of Extended Data Figure 8.

Given the KDE is not mentioned until the last paragraph of the results section and is reliant on the identification of phylogeographic outliers, it would make more sense to have the KDE visualisation currently in Fig 1 parts C and D as part of Fig 4.

Response: We would prefer to keep the Figures 1C and D as part of Figure 1 to allow the reader to follow the ‘process’ of the analysis and directly compare the different versions of the maps with each other. Furthermore, Figure 5 (formerly Figure 4) is showing a completely different dataset, so that moving both maps to the end of the manuscript would rather require a new Figure 6. However, with now 5 figures and 1 table we have already reached the limit of elements for the main manuscript.

Further issues/queries associated with Fig 4

The significance of Y having brackets (while N and F do not) is not stated in the legend.

It therefore does not make sense why is Y described in the legend while N and F are

not.

‘Sequence Z21_0139 (Y) is marked as a potential outlier due to its close genetic relation to animal sequences in more than 200 km distance’ - This is confusing as having noted the criteria for outlier definition, I would have taken from the paper that more accurately, it is the absence of geographically close genetic relations rather than the presence of geographically distant close genetic relations that the described method for outlier detection relies on. However, I think the point is that Y would not be classified as an outlier using the definition elsewhere in the paper - It is not clear from the legend that the standard definition is being overridden.

Response: The two cases (Y) and (Z) were individually marked to show, that they possibly represent infections at aberrant locations, despite not meeting our outlier criteria. The suspicion of an aberrant infection source is based on the observation, that their closest relatives (in both cases being almost identical) are located far from their location. However, other less closely related sequences with >98.6% nt identity are located at <37.9km distance.

We agree, that calling these two cases ‘potential outliers’ may be confusing and give the impression that we are overriding our own criteria. This is not the case, since only the ‘real’ outliers were treated as outliers for further analyses (e.g. KDE), while these two cases were treated like any other ‘non-outlier’. We have therefore now labelled these cases as #1 and #2 and call them ‘additional cases with potentially aberrant infection sources’ throughout the manuscript. They have been explained in more detail in a separate Extended Data Table 4 and we emphasize that this is only an informative labelling without consequences for subsequent analyses.

Were there any instances where a human BoDV-1 sequence had >1 sequence with equally minimal patristic distance? If so, to which of these would the spatial distance be plotted in Fig 4?

Response: Yes, some human sequences possessed equal patristic distances (or almost equal patristic distances, defined as a value within 1.2-fold the minimal patristic distance) to two or more sequences. In these cases, the distance of the spatially closest of these two or more cases was used. This information is provided in the figure legend (lines 1097-1099).

There is a green point at around 0.010 on the x-axis that appears above the dashed line at 39.8km. Why is this not identified as an outlier using the stated 37.9km cutoff?

Similarly, it is confusing to see F identified as a phylogeographic outlier when it falls on the median line. The reader currently must jump around the manuscript somewhat to figure out why this is the case, which is not ideal.

Response: The outlier definition and the data shown in Figure 4 are based on two entirely different data sets and analysis procedures. While the outlier definition is based on the minimal spatial distance to ANY other sequence with $\geq 98.6\%$ sequence identity, Figure 4 shows the spatial distance of the human-derived sequences to ONLY their closest relatives (based on patristic distances and regardless if these had $\geq 98.6\%$ identity or less, as for outlier F, and regardless if other sequences with $\geq 98.6\%$ identity to the respective human case were located at closer distance than the next relatives). The position of outlier F is below 37.9km (which is by chance close to the 90th percentile of this dataset) since its closest sampled relative is located in rather close proximity. However, since this closest relative is only very distantly related (98.1% nt identity), case F nevertheless fulfills the criteria of our outlier definition. Other human sequences (such as the green point mentioned by the reviewer or case #1, previously labelled as ‘potential outlier Y’) may be located at a distance of $>37.9\text{km}$ to their closest relatives, so that they are plotted above 37.9km here, but additionally may have other related sequences ($\geq 98.6\%$ identity) located within 37.9km radius, thus not being classified as outliers.

Minor

Line 60: clarify wild shrews?

Response: We added the word ‘wild’.

Line 85-86. ‘Within days to months’ relative to what.. exposure, onset of fever, hospitalisation?

Response: We have added the information, that this is relative to the onset of neurological symptoms.

Line 88. Reference 15 duplicated

Response: Thank you for pointing this out. The duplicate was deleted.

Line 110. Shrew sample origins could be briefly outlined.

Response: The origin of the shrew samples has been added.

Line 258-259. Substitution model components should be written out before acronyms at first use. ‘General time reversible model of nucleotide substitution’.. ‘a proportion of invariant sites’ etc. I’m not familiar with ‘F’ in the N-X/P model, does that denote estimated base frequencies?

Response: The provided model names are those used in the mentioned software. We provided a more detailed text to explain the model names.

Line 260-261. Not clear what was done with ggtree here. Is that simply the exclusion of the outgroup sequence for visualisation?

Response: Ggtree was exclusively used to visualize the phylogenetic tree without the outgroup (Figures 2 and 3) and to visualize the single phylogenetic clusters and subclusters separately in the extended data Figure 5. We have specified this in the text (line 661 to 664).

Ideally, the tree including the outgroup should be included in the supplementary figures.

Response: We added the raw tree files in Newick format to the supplemental material.

Line 266-267. Please state reason for exclusion of laboratory strains in the text. Was this because dates were unknown? Or to guard against lab-adaptative mutations that may have been acquired since original sampling date?

Response: We have added the information that they were excluded to guard against lab-adaptive mutations (lines 670-671).

Line 340. 'Clusters 1 to 4 and subclusters 1A and 1B' would be more accurate. The presence of subclusters within it does not negate the presence of cluster 1.

Response: We agree with the reviewer and have modified the text accordingly (line 180).

REVIEWERS' COMMENTS

Reviewer #2 (Remarks to the Author):

The authors have worked and considered my suggested revisions and those of the co-reviewers, and I am happy to see this work published. No further comments.

Reviewer #3 (Remarks to the Author):

I really appreciate the care with which the authors have responded to the comments of both myself and the other reviewers. In particular, I think there are some very helpful small but precise changes surrounding the definition and references to phylogeographic outliers, such as the upfront reference to F around line 259 and changing how cases now labelled #1 and #2 are referred to. Some small changes throughout the manuscript really help interpretability. The addition of the analysis of patristic distances is also very nice to see for comparison.

A couple of very minor points to be addressed:

Figure legends. Please change 'Maximum likelihood' (Figs 2, 4) and 'Maximum Likelihood' (Fig 3) to 'maximum likelihood'.

Figure 5. Please change the the slope from '2,544' to '2.544' (displaying to 4 significant figures is also probably overkill, not sure if there is any belief that the estimate from these data is so accurate).

Reviewer #2 (Remarks to the Author):

The authors have worked and considered my suggested revisions and those of the co-reviewers, and I am happy to see this work published. No further comments.

Reviewer #3 (Remarks to the Author):

I really appreciate the care with which the authors have responded to the comments of both myself and the other reviewers. In particular, I think there are some very helpful small but precise changes surrounding the definition and references to phylogeographic outliers, such as the upfront reference to F around line 259 and changing how cases now labelled #1 and #2 are referred to. Some small changes throughout the manuscript really help interpretability. The addition of the analysis of patristic distances is also very nice to see for comparison.

A couple of very minor points to be addressed:

Figure legends. Please change 'Maximum likelihood' (Figs 2, 4) and 'Maximum Likelihood' (Fig 3) to 'maximum likelihood'.

Response: Maximum Likelihood' was changed to 'maximum likelihood' throughout the manuscript and supplementary files.

Figure 5. Please change the the slope from '2,544' to '2.544' (displaying to 4 significant figures is also probably overkill, not sure if there is any belief that the estimate from these data is so accurate).

Response: The decimal marker of the slope in figure 5 was corrected. The number of digits was not changed to keep it consistent with the other figures of the manuscript.